# Subtilase SBT5.2 inactivates flagellin immunogenicity in the plant apoplast

Pierre Buscaill [1,4], Nattapong Sanguankiattichai[1], Farnusch Kaschani [2], Jie Huang[1], Brian C. Mooney[1], Yuge Li[1,3], Joy Lyu[1], Daniela Sueldo[1,5], Markus Kaiser [2] & Renier A. L. van der Hoorn [1] ✉

Most angiosperm plants recognise the 22-residue flagellin (flg22) epitope in bacterial flagellin via homologs of cell surface receptor FLS2 (flagellin sensitive-2) and mount pattern-triggered immune responses. However, flg22 is buried within the flagellin protein indicating that proteases might be required for flg22 release. Here, we demonstrate the extracellular subtilase SBT5.2 not only releases flg22, but also inactivates the immunogenicity of flagellin and flg22 by cleaving within the flg22 epitope, consistent with previous reports that flg22 is unstable in the apoplast. The prolonged lifetime of flg22 in *sbt5.2* mutant plants results in increased bacterial immunity in priming assays, indicating that SBT5.2 counterbalances flagellin immunogenicity to provide spatial-temporal control and restrict costly immune responses and that bacteria take advantage of the host proteolytic machinery to avoid detection by flagellin having a protease-sensitive flg22 epitope.

Bacterial flagellin is a strong inducer of innate immune responses in both plants and animals. The flagellum enables bacterial motility and is a polymer that consists of thousands of flagellin proteins in a tubular assembly[1–4]. The surface of the flagellum is glycosylated with *O*-glycans that can differ in composition between bacterial strains[5].

In most angiosperms, the recognition of bacterial flagellin is mediated by a cell surface-localised receptor kinase called FLS2 (flagellin sensing-2[6]). FLS2 recognises a highly conserved epitope of 22 amino acids in the N-terminal region of flagellin, called flg22[7,8]. The flg22 peptide is routinely used to trigger pattern-triggered immunity (PTI), but how flg22 is made available during infection is not yet clear.

It has been predicted that to expose flg22, the flagellin protein needs to be processed from its monomeric precursor because the flg22 epitope is not fully exposed in the flagellin monomer, and is unlikely to bind the FLS2 receptor in this conformation[9]. Indeed, a protease inhibitor cocktail can suppress the perception of flagellin, but not of flg22[10]. A recent report describes that Arabidopsis subtilases SBT5.2 and SBT1.7 are processing flagellin at the C-terminus of flg22[11].

However, despite the reduced processing of flagellin in *sbt5.2/sbt1.7* mutants, these mutants only show a 2-minute delay in the maximum release of reactive oxygen species (ROS). Although a hampered flagellin perception should have caused an increased bacterial susceptibility, increased bacterial growth on *sbt5.2/sbt1.7* mutants has not been reported.

Here, we describe how flagellin is quickly inactivated in the apoplast of the model plant *Nicotiana benthamiana* by subtilase SBT5.2 and other proteases. In addition to processing flg22 from its precursor, SBT5.2 quickly inactivates the immunogenic activity of flagellin by cleaving in the middle of the flg22 epitope. Indeed, *sbt5.2* mutants show increased stability of flagellin and flg22 in the apoplast, associated with prolonged immunogenicity. Consequently, *sbt5.2* mutants are less susceptible to bacterial colonisation when primed with low flg22 concentrations, indicating that SBT5.2 provides spatial-temporal control over immune responses to restrict the induction of costly defence responses and that pathogens take advantage of the host proteolytic machinery by exposing protease-sensitive epitopes.

[1]The Plant Chemetics Laboratory, Department of Biology, University of Oxford, Oxford, UK. [2]ZMB Chemical Biology, Faculty of Biology, University of Duisburg-Essen, Essen, Germany. [3]Guangdong Provincial Key Laboratory of Applied Botany and State Key Laboratory of Plant Diversity and Prominent Crops, South China Botanical Garden, Chinese Academy of Sciences, Guangzhou, China. [4]Present address: School of Biological Sciences, University of Bristol, Bristol, UK. [5]Present address: Department of Biology, University of Science and Technology, Trondheim, Norway. ✉e-mail: renier.vanderhoorn@biology.ox.ac.uk

## Results

### Flagellin monomer is immunogenic, but its polymer is not

To study the processing of flagellin in the apoplast, we isolated flagella from *Pseudomonas syringae* pv. *tabaci* 6605 (*Pta*6605) to obtain polymeric flagellin that we monomerized by heat-treatment (10 min. 70 °C). To confirm the monomerization, we used a 100 kDa molecular weight cut-off filter (MWCO). Polymeric flagellin was unable to pass the 100 kDa MWCO filter, in contrast to monomeric flagellin (Fig. 1a). We next tested if polymeric flagellin (sample a) and monomeric flagellin (sample b) could trigger an oxidative burst in leaf discs from 4-week-old *Nicotiana benthamiana* floating on a solution containing luminol and horse radish peroxidase (HRP). Only monomeric flagellin triggered a burst of reactive oxygen species (ROS) and polymeric flagellin did not (Fig. 1b), demonstrating that only monomeric flagellin is recognised. Furthermore, monomerised flagellin did not trigger an oxidative burst in leaf discs of *fls2* mutant *N. benthamiana* plants[12] (Fig. 1c), demonstrating that the detected oxidative burst is caused by flagellin recognition via its FLS2 receptor and that no other elicitors are present in the purified flagellin samples.

### Flagellin monomer is quickly degraded in AF but its polymer not

To investigate flagellin processing by apoplastic proteases, we incubated the flagellin polymer and monomerised flagellin with apoplastic fluid (AF) isolated from *N. benthamiana* and separated the proteins on protein gels to monitor the degradation of flagellin in AF over time. Monomeric flagellin rapidly degrades in AF within minutes, whereas polymeric flagellin remains stable in AF for over 24 h (Fig. 1d), consistent with the hypothesis that polymeric flagellin is protected from degradation. ROS assays with these samples revealed that the oxidative burst quickly disappears when flagellin is incubated with AF for 30 or 60 min (Fig. 1e), indicating that the immunogenicity of monomerised flagellin quickly disappears in the apoplast.

### Flagellin is degraded by apoplastic proteases

To further investigate flagellin processing in AF, we precipitated the proteins with 80% acetone and analysed the peptides in the supernatant by liquid chromatography-tandem mass spectrometry (LC-MS/MS) to detect the peptides released from flagellin when incubated in AF[10]. We identified a total of 466 different flagellin-derived peptides, spanning the entire protein sequence except for peptides containing the six *O*-glycosylation sites of flagellin (Supplementary Data 1 and Fig. 2a), indicating that monomeric flagellin is not fully deglycosylated before it is processed into peptides. Indeed, peptides covering this region were detected upon incubation of nonglycosylated flagellin from the flagellin glycosyl transferase mutant *Δfgt1*[13] when incubated in AF (Supplementary Data 2 and Supplementary Fig. 1),

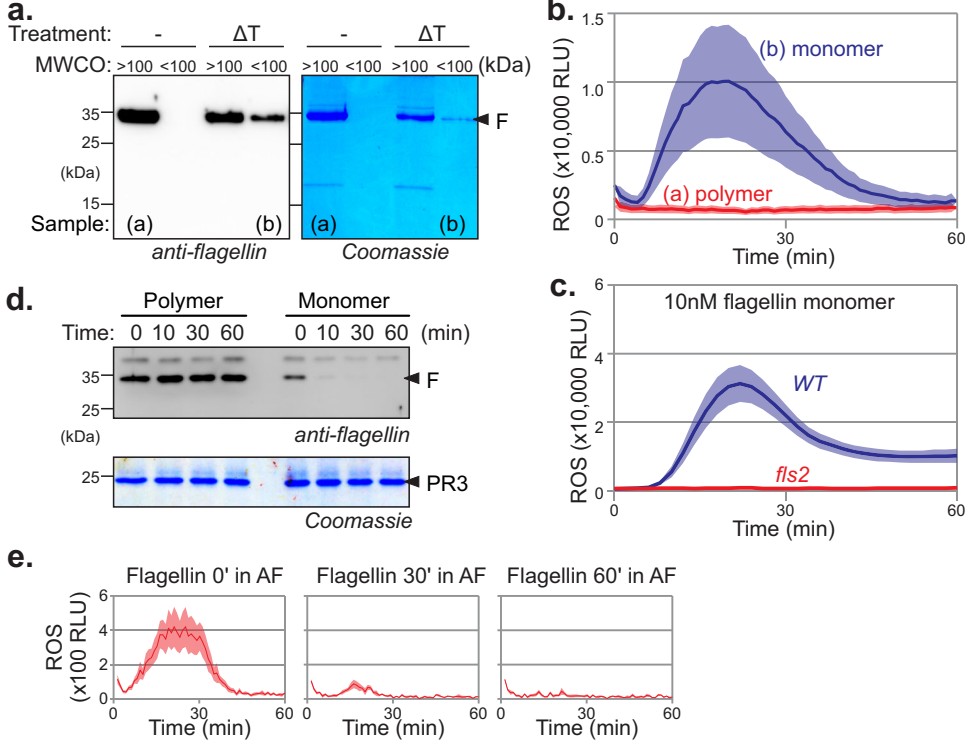

**Fig. 1 | Monomeric flagellin is recognised and quickly inactivated extracellularly. a** Flagellin monomerisation. Purified flagellin from *Pta*6605 bacteria was treated with/without heat (10′ 70 °C, ΔT) and both samples were centrifugated through a 100 kDa MW cut-off filter. The flow-through (<100 kDa) and the residue (>100 kDa) for each sample were incubated for 30 min and separated on 15% SDS-PAGE and stained with Coomassie or analysed by western blotting using anti-flagellin antibodies. **b** Monomeric but not polymeric flagellin is recognised in *N. benthamiana*. Leaf discs of wild-type *N. benthamiana* floating on a solution containing luminol and horse radish peroxidase (HRP) were incubated with 10 μg/mL flagellin proteins (samples a and b from Fig. 1a) and the luminescence was measured over time. Lines represent the mean, and error shades represent the SE of $n = 6$ replicates. RLU, relative luminescence units. **c** Mutant *fls2* is not responding to monomerised flagellin. Leaf discs of wild-type and *fls2* mutant *N. benthamiana* floating on a solution containing luminol and horse radish peroxidase (HRP) were incubated with 10 μg/mL monomerised flagellin and the luminescence was measured over time. Lines represent the mean, and error shades represent the SE of $n = 6$ replicates. RLU, relative luminescence units. **d** Monomeric flagellin but not polymeric flagellin is quickly degraded in the apoplast of *N. benthamiana*. Purified polymeric flagellin was monomerised by heat treatment and incubated with apoplastic fluids (AF) isolated from *N. benthamiana* leaves for 0, 10, 30 and 60 min and analysed by western blot using anti-flagellin antibodies and by Coomassie staining. Pathogenesis-related-3 (PR3) is an abundant apoplastic protein and served here as a loading control. **e** Monomerised flagellin is quickly inactivated in AF. 10 μg/ml monomeric flagellin was incubated with AF for 0, 30 or 60 min, and treated samples were added to leaf discs floating on luminol-HRP and ROS release was monitored using a luminescence plate reader for 60 min. Lines represent mean, and error shades represent SE of $n = 6$ biological replicates. RLU, relative luminescence units.

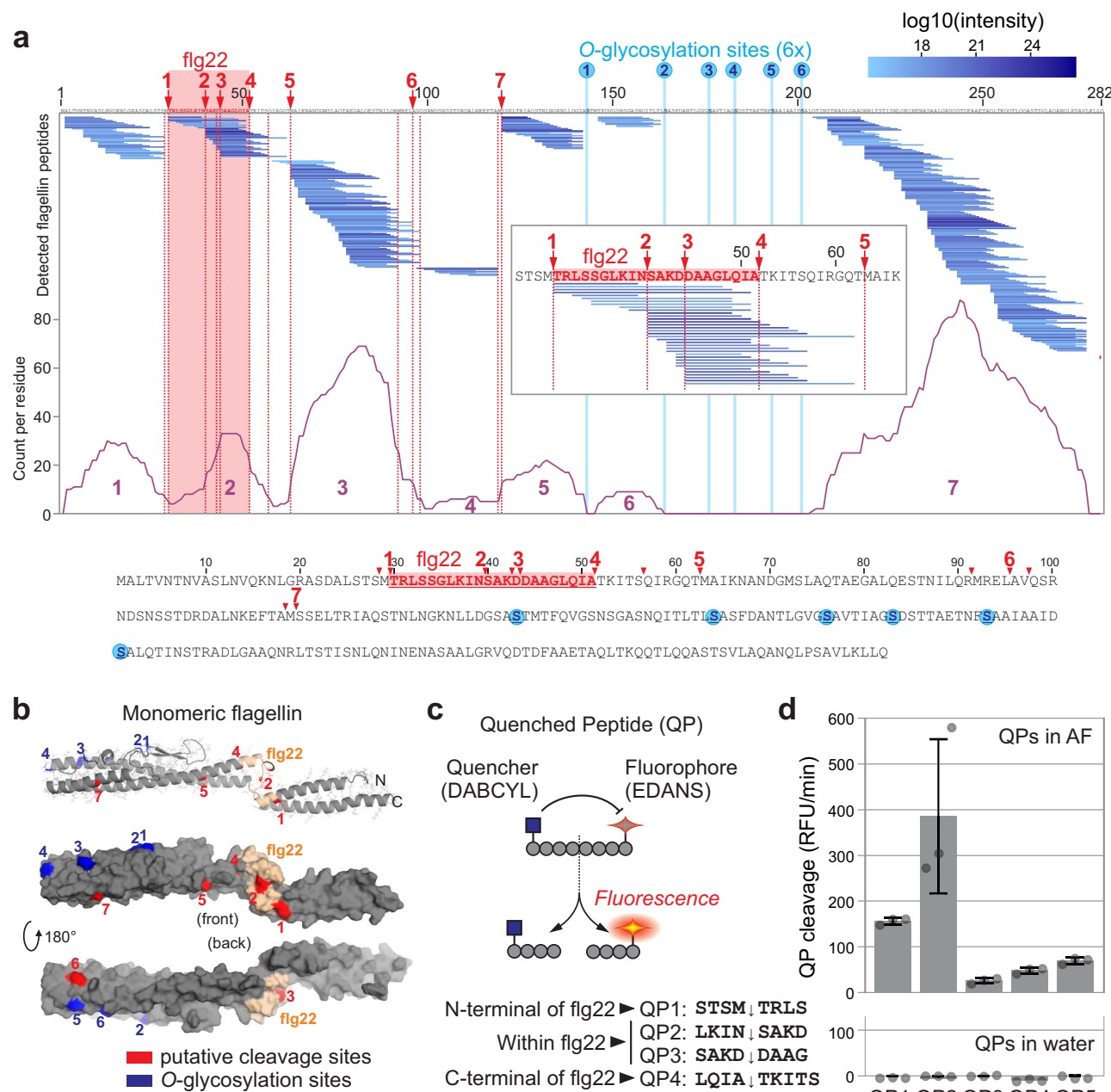

**Fig. 2 | Flagellin is degraded by apoplastic proteases. a** Degradation products of flagellin in AF detected by LC-MS/MS analysis. Purified flagellin was incubated with AF isolated from *N. benthamiana* leaves. Proteins were precipitated with 80% acetone and the supernatant (peptide fraction) was analysed by LC-MS/MS. Flagellin-derived peptides were matched to the flagellin protein sequence. Highlighted are the flg22 sequence (red), putative cleavage sites (red dashed lines) and six *O*-glycosylation sites (light blue lines). The number of times each residue was detected in the peptides is indicated in the purple graph, showing the peptides aligned in seven clusters. Below: position of the putative cleavage sites within the flagellin protein sequence. Highlighted are flg22 (red), putative cleavage sites (red and the six *O*-glycosylation sites (blue). Inset: region containing flg22 and cleavage sites 1–5 with the corresponding detected peptides. **b** Location of putative cleavage sites within the predicted structure of the flagellin monomer, generated with AlphaFold. Highlighted are the putative cleavage sites (red), and the *O*-glycosylation sites (blue). **c** Concept of using quenched peptides (QPs) to monitor processing. QP cleavage separates the fluorophore from the quencher, resulting in fluorescence. Five QPs containing the cleavage regions in flagellin were custom-synthesised. **d** All QPs are processed in AF but with different efficiencies. QPs were mixed with water or AF and fluorescence was measured immediately over 5 min. Bars represent mean, and error bars represent SE of *n* = 3 different AFs.

The detected peptides released from *Pta*6605 flagellin assemble into seven clusters (Fig. 2a, purple line). This peptide distribution pattern suggests that flagellin undergoes cleavage at specific sites by endopeptidases and that exopeptidases subsequently remove residues from the N- and C-termini.

Closer inspection of the protein coverage indicates that flagellin might be cleaved by endopeptidase(s) at LSTS ↓ M ↓ TRLS, LQIA ↓ TKITS ↓ QIRGQT ↓ MAIK, ILQR ↓ MREL ↓ AV ↓ QSRN and EFRA ↓ M ↓

SSEL (Fig. 2a), of which the first two processing sites would release flg22 from its precursor.

Because we detected a loss of immunogenicity of flagellin, we also searched for putative processing sites within the flg22 sequence and found two candidate regions: LKIN ↓ SAKD and SAKD ↓ DAAG (Fig. 2b, inset). Processing in either of these sites would inactivate the elicitor activity[7]. Collectively, the putative cleavage sites do not have an obvious consensus sequence.

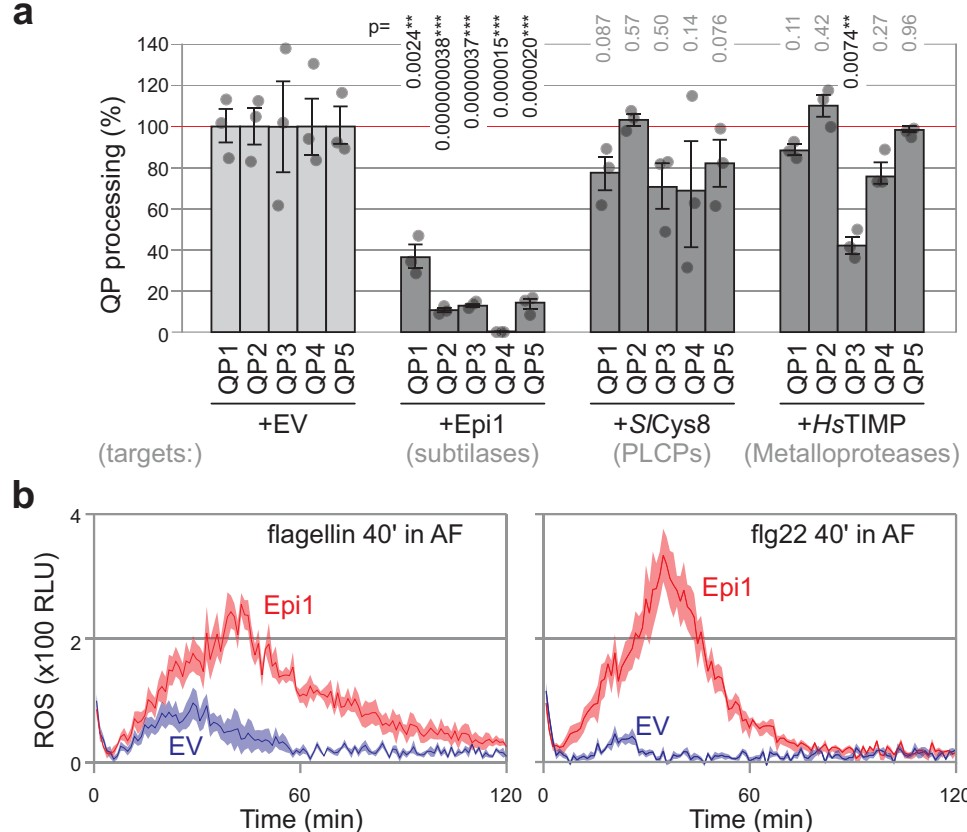

**Fig. 3 | Subtilase-inhibitor Epi1 suppresses processing of Qps, flg22 and flagellin. a** Subtilase inhibitor, Epi1 suppresses the processing of all quenched peptides (QPs). Apoplastic fluids (AFs) isolated from *N. benthamiana* leaves transiently expressing protease inhibitors Epi1, *Sl*Cys8 and *Hs*TIMP were mixed with QPs and fluorescence and were immediately monitored using a plate. QP processing in the empty vector (EV) control was set at 100% for each QP. Bars represent mean, and error bars represent ± SE of n = 3 biological replicates (i.e., AFs from 3 plants). Two-sided Students *t* test: **p* < 0.05; ***p* < 0.01; ****p* < 0.001. **b** Epi1 increases the immunogenicity of monomeric flagellin and flg22. AFs from plants transiently expressing Epi1 or the empty vector (EV) control were incubated with 10 µg/ml monomeric flagellin or 100 nM flg22 for 40 min and then added to leaf discs of *N. benthamiana* plants floating on luminol and horse radish peroxidase (HRP). ROS release was measured for 120 min by luminescence (relative light units, RLU) using a plate reader. Lines represent mean, and error shades represent SE of n = 6 replicates. RLU, relative luminescence units.

We next mapped the putative cleavage sites onto a model of the structure of the flagellin monomer. The putative cleavage sites distribute over the hinge region and various positions in α-helixes that are not exposed in polymeric glycosylated flagellin (Fig. 2b), explaining why monomeric flagellin is more susceptible to proteolysis than polymeric flagellin.

To confirm the cleavage events within flagellin, we designed and custom-synthesised quenched peptides (QPs) containing different cleavage sites, flanked by an N-terminal quencher (DABCYL) and a C-terminal fluorophore (Glu-EDANS). Processing these peptides releases a fluorophore that is detected by fluorescence (Fig. 2c). The QPs cover the two sites within flg22 (LKIN ↓ SAKD (QP2) and SAKD ↓ DAAG (QP3)); the flg22-flanking sites STSM ↓ TRLS (QP1) and LQIA ↓ TKITS (QP4), as well as one additional obvious putative processing site following the flg22 sequence (RGQT ↓ MAIK (QP5)). Incubation of these QPs in AF causes a rapid increase in fluorescence when compared to the water control (Fig. 2d), indicating that these peptides are cleaved in AF. These data also indicate that QP2 is most efficiently cleaved in AF compared to the other four QPs.

### Flagellin is processed by apoplastic subtilases

To identify the class of proteases responsible for processing flagellin, we isolated AFs from leaves overexpressing protease inhibitors Epi1, *Sl*Cys8 and *Hs*TIMP, which inhibit subtilases (SBTs), papain-like Cys proteases (PLCPs) and metalloproteases, respectively[14–16]. Incubation

of these AFs with quenched peptides revealed that Epi1 blocks processing of all QPs, whereas *Hs*TIMP only suppresses QP3 processing and *Sl*Cys8 and has no effect (Fig. 3a). QP1 processing is less efficiently blocked by Epi1, indicating that this site is cleaved by a protease different from an Epi1-sensitive subtilase. The near complete blockage of QP4 processing by Epi1 indicates that only subtilases cleave at this region. Processing of QP3 is suppressed more by Epi1 than by *Hs*TIMP, again indicating that QP3 is processed by subtilases rather than other proteases. These results indicate that subtilases are responsible for most of the flagellin processing in AF.

We next tested if Epi1 can reduce the inactivation of immunogenicity in AF by incubating flagellin or flg22 in AF for 40 min and measuring their ability to trigger the oxidative burst. Importantly, immunogenic activity was detected after incubation of flagellin or flg22 peptide with AFs isolated from leaves transiently expressing Epi1, but not from the EV control (Fig. 3b), indicating that subtilases are responsible for degrading the immunogenicity of both flagellin and flg22.

### Flagellin processing requires SBT5.2 subtilases

The *N. benthamiana* genome encodes for 72 putative subtilases[17,18]. Among the ten subtilases detected in the apoplast, four are notably more abundant: SBT5.2a, SBT1.9a, SBT1.7a, and SBT1.7c[15,18,19], also referred to as SBT1-4, respectively[20]. To identify the subtilase responsible for flagellin processing, we depleted transcripts of these four

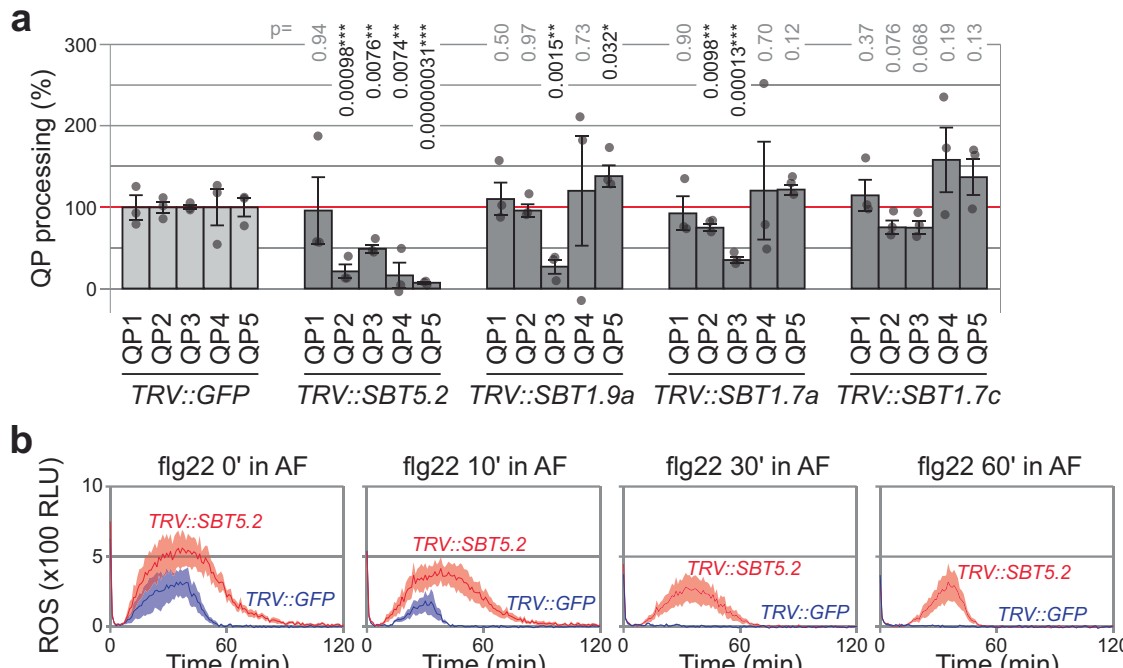

**Fig. 4 | SBT5.2 is required for processing QPs 2-5 and inactivating flg22. a** SBT5.2 is required for QP processing in AF. AF from plants silenced for various subtilases was mixed with 10 μM quenched peptides, and fluorescence was monitored immediately using a plate reader. QP processing in AF of *TRV::GFP* control plants was set at 100%. Bars represent mean, and error bars represent SE of *n* = 3 different plants. Two-sided Students *t* test: *\*p* < 0.05; \*\**p* < 0.01; \*\*\**p* < 0.001. Please note that due to variation in silencing efficiency and the low number of replicates, QP1

processing in *TRV::SBT5.2* plants might be disturbed by an outlier, and the significance of reduced QP2 processing in SBT1.7a plants may not be relevant. **b** *SBT5.2* silencing reduces the degradation of flg22 immunogenicity. 100 nM flg22 was incubated in AF from *TRV::SBT5.2* or *TRV::GFP* plants for 0, 30 and 60 min before adding to leaf discs from *N. benthamiana* floating in HRP or luminol. Luminescence was measured and plotted against time. Lines represent mean, and error shades indicate SE of *n* = 6 replicates. RLU, relative luminescence units.

subtilases using Virus-Induced Gene Silencing (VIGS) with Tobacco Rattle Virus (TRV) vectors carrying 300 bp fragments targeting these subtilases. Each fragment was designed to target one subtilase, except for the fragment present in *TRV::SBT5.2*, which targets the three *SBT5.2* homologues (*a-c*) for which transcripts were detected in *N. benthamiana*[18] (Supplementary Fig. 2).

To determine which subtilases are required for processing the quenched peptides, QPs were incubated with AFs isolated from *TRV::SBT* plants and fluorescence was measured. Processing of all QPs except QP1 was significantly reduced in the AF of *TRV::SBT5.2* plants compared to the *TRV::GFP* control (Fig. 4a). The absence of reduced QP1 processing is consistent with the observation that QP1 processing is also less sensitive to inhibition by Epi1 and indicates that QP1 is not processed by abundant, Epi1-sensitive subtilases. QP3 processing is not as strongly reduced in *TRV::SBT5.2* plants compared to the other QPs, whereas QP3 processing was also reduced in AF isolated from *TRV::SBT1.9a* and *TRV::SBT1.7a* plants. This is consistent with SBT1.9a being a phytaspase[17,21], which prefers cleaving after aspartic residues present in QP3 (SAKDDAAG). Taken together, these results indicate that SBT5.2 s are responsible for processing QPs 2–5, but QP3 is also cleaved by phytaspase SBT1.9a and SBT1.7a.

Since SBT5.2 s seem responsible for processing flg22 at sites 2 and 3, we tested the stability of flg22 immunogenicity by incubating flg22 various times in AF isolated from *TRV::SBT5.2* and *TRV::GFP* plants. Importantly, the decline of flg22 immunogenicity is much slower in AF of *TRV::SBT5.2* plants compared to *TRV::GFP* plants (Fig. 4b), indicating that SBT5.2 s are responsible for inactivating flg22.

**Flagellin/flg22 is stabilised in AF of *sbt5.2* plants**
To further investigate the role of SBT5.2 in flagellin processing, we took advantage of two recently described *sbt5.2* triple mutant lines of *N. benthamiana*, which are disrupted in the open reading frames

encoding all three *SBT5.2* genes[18]. Activity-based profiling with FP-TAMRA on AF demonstrates that the *sbt5.2* mutants lack the most active subtilases at 70 kDa (Fig. 5a), consistent with being the most active subtilase in AF of *N. benthamiana*[15,19,20].

Processing of QP1, QP2 and QP5 is significantly reduced in AF of *sbt5.2* mutant plants (Fig. 5b). QP3 processing is only slightly reduced in AF of *sbt5.2* mutant plants, whereas QP4 processing is strongly reduced but not significant (Fig. 5b), However, all QPs are still cleaved to some extent in the absence of *sbt5.2*, suggesting that other proteases in AF might slowly process these peptides in the absence of SBT5.2 s.

Consistent with flagellin protein degradation, the flg22 peptide retains immunogenicity significantly longer when incubated in AF isolated from *sbt5.2* plants, whereas flg22 immunogenicity is quickly lost when incubated with AF of WT plants (Fig. 5c).

Incubation of monomerised flagellin with AF isolated from WT and *sbt5.2* mutant plants revealed that flagellin was more stable in the AF of both *sbt5.2* mutants than in the AF of WT plants (Fig. 5d), demonstrating robustly that SBT5.2 s are necessary for flagellin processing. Interestingly, in addition to the full-length flagellin protein we also detect the accumulation of several intermediate flagellin degradation products in the AF of *sbt5.2* plants, suggesting that other proteases slowly process flagellin in the absence of SBT5.2 s.

We next monitored the timing of the oxidative burst triggered by flagellin monomers in leaf discs of WT and *sbt5.2* mutants, to compare to the recently reported 2-minute delay in the peak of the ROS burst in the Arabidopsis *sbt5.2/sbt1.7* mutant[11]. In our assay, however, both the timing and the amplitude of the oxidative burst upon adding flagellin are similar between WT and *sbt5.2* mutant plants (Fig. 5e). Statistical analysis of the peak times did not reveal a significant difference between WT and *sbt5.2* mutant *N. benthamiana* (Fig. 5f). These data indicate that flagellin processing by SBT5.2 does not affect flagellin

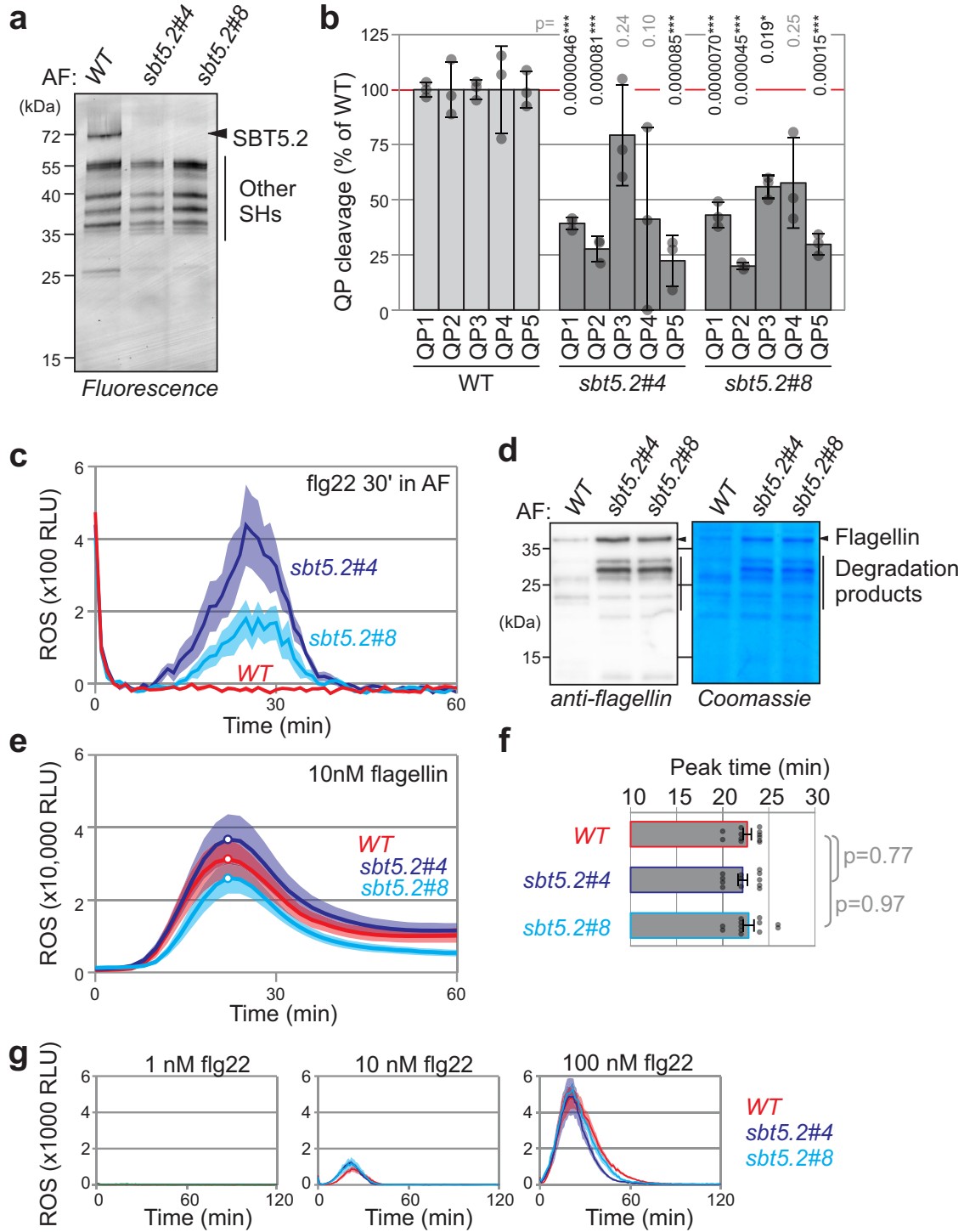

recognition. ROS responses to various flg22 concentrations are also indistinguishable between leaf discs of WT and *sbt5.2* mutant *N. benthamiana* (Fig. 5g and Supplementary Fig. 3), indicating that when added to leaf discs, flg22 is not inactivated by SBT5.2, possibly because the SBT5.2 is diluted in the medium.

## Purified SBT5.2 processes flagellin and flg22

To determine if SBT5.2 is sufficient for processing QPs and flagellin and inactivating flg22, we purified SBT5.2a-His from agroinfiltrated plants using Nickel-NTA agarose[22], and used activity-based labelling with FP-TAMRA to confirm that this is active (Supplementary Fig. 4a). Transiently expressed secreted GFP-His was included as a negative control

(Supplementary Fig. 4b). Incubation of these purified proteins with the QPs revealed that all QPs are substrates of SBT5.2a-His, and that QP2 is the best substrate and that QP3 is hardly cleaved, whereas no substrate is cleaved in the GFP-His control (Fig. 6a). Furthermore, incubation of flg22 with purified SBT5.2a-His inactivated its elicitor activity (Fig. 6b). Incubation of monomerised flagellin with purified SBT5.2-His caused a quick degradation of flagellin in contrast to the purified GFP-His control (Fig. 6c and Supplementary Fig. 5). The peptides released from flagellin by purified SBT5.2-His were identified by LC-MS/MS and confirm that SBT5.2a-His cleaves at site-1, -2 and -4-7 but not at site-3 (Fig. 6d and Supplementary Data 3). The few flagellin-derived peptides detected in the GFP-His control were also detected upon incubation of

**Fig. 5 | *sbt5.2* mutants show reduced flagellin and flg22 processing. a** SBT5.2 s are the major active subtilases in the apoplast. AF was isolated from WT plants, and two triple *sbt5.2* mutants and serine hydrolases (SHs) were labelled with FP-TAMRA and detected by in-gel fluorescence scanning. **b** QP processing is reduced in AF isolated from both *sbt5.2* triple mutants. 10 μM QPs were incubated with AF of WT or *sbt5.2* mutants and fluorescence was measured with a plate reader. Bars represent mean, and error bars represent SE of *n* = 4 samples. Two-sided Students *t* test: **p* < 0.05; ***p* < 0.01; ****p* < 0.001. **c** flg22 inactivation is reduced in AF isolated from both *sbt5.2* triple mutants. AF from leaves of WT or *sbt5.2* mutant plants were incubated with 100 nM flg22 peptide for 30 min then added to *N. benthamiana* leaf discs floating on luminol-HRP solution. The ROS burst was monitored using a luminescence plate reader. Lines represent mean, and error shades represent SE of *n* = 6 biological replicates. RLU, relative luminescence units. **d** Reduced flagellin processing in the *sbt5.2* mutants. Purified flagellin was incubated for 30 min with AF

from leaves of WT or *sbt5.2* mutants. The samples were separated on SDS-PAGE and stained with Coomassie or analysed by western blotting using anti-flagellin antibodies. **e** The timing of the oxidative burst upon flagellin recognition is indistinguishable between WT plants and *sbt5.2* mutants. 10 nM flg22 was added to *N. benthamiana* leaf discs floating on the luminol-HRP solution. The ROS burst was monitored using a luminescence plate reader. Lines represent mean, and error shades represent SE of *n* = 12 biological replicates. RLU, relative luminescence units. **f** Peak times taken from (**e**) are not different between WT and *sbt5.2* mutants. Bars represent mean, and error bars represent SE of *n* = 12 replicates and *p*-values were calculated with ANOVA by two-sided Tukey HSD test. **g** No altered response to flg22 in *sbt5.2* mutants. Leaf discs from 4-week-old *N. benthamiana* WT or *sbt5.2* mutants floating on luminal-HRP were treated with 1, 10 or 100 nM flg22 and ROS burst was monitored using a plate reader. Lines represent mean, and error shades represent SE of *n* = 6 biological replicates.

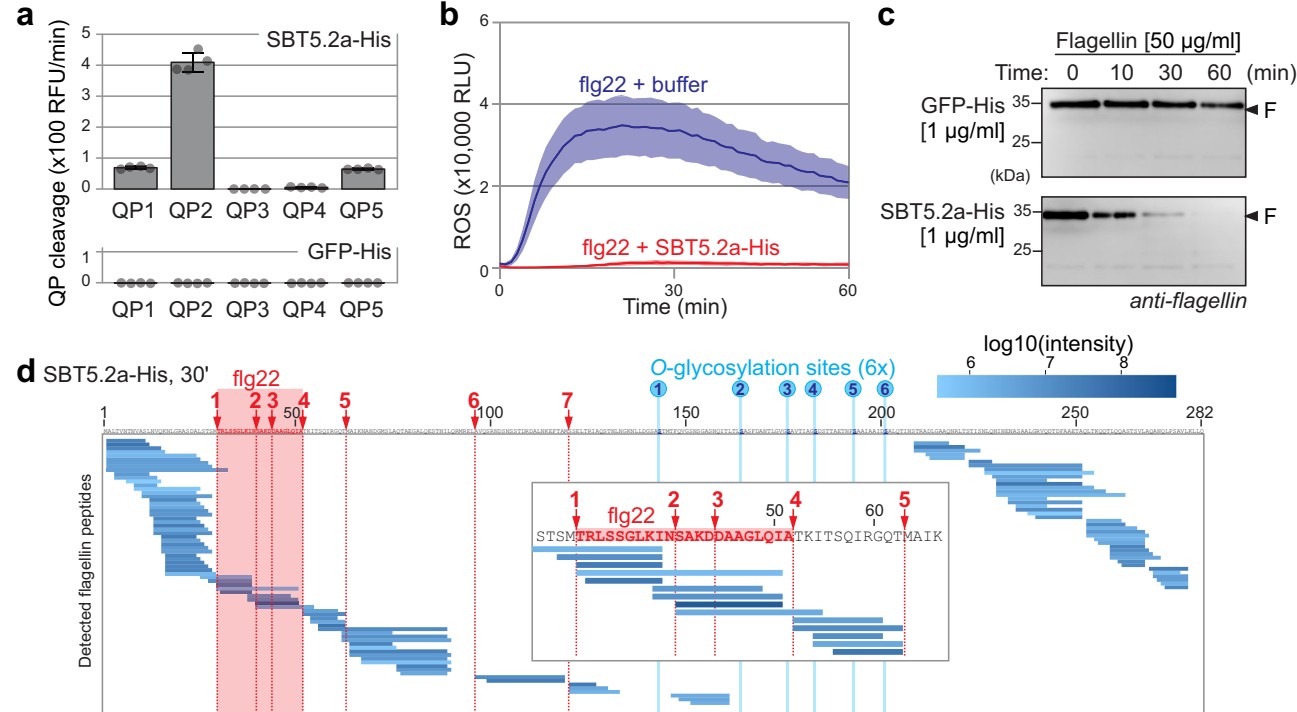

**Fig. 6 | Purified SBT5.2a degrades flagellin and inactivates flg22. a** Purified SBT5.2a-His cleaves QPs. QPs were incubated with 10 μl of 25 μg/ml purified SBT5.2a-His or GFP-His and fluorescence was measured. Bars represent mean, and error bars represent SE of *n* = 4 replicates. **b** Purified SBT5.2a-His inactivates the flg22 elicitor. 50 nM flg22 was incubated with 10 μg/ml purified SBT5.2a-His for 40 min. These samples were added to leaf discs of *N. benthamiana* plants floating on the luminol-HRP solution, and the ROS burst was monitored using a luminescence plate reader. Lines represent mean, and error shades represent SE of *n* = 6

biological replicates. RLU, relative luminescence units. **c** Purified SBT5.2a-His quickly processes flagellin. Purified flagellin (50 μg/ml) was incubated with purified SBT5.2a-His or GFP-His (1 μg/ml) for 0, 10, 30 or 60 min and samples were separated on SDS-PAGE and stained with Coomassie or analysed by western blotting using anti-flagellin antibodies. An independent replicate is shown in Supplemental Fig. 5b. **d** Purified SBT5.2a-His cleaves flagellin in the flg22 epitope. Flagellin was incubated with purified SBT5.2-His for 30 min and the released peptides were analysed by LC-MS/MS. Shown is the mean of *n* = 3 replicates.

flagellin in water (Supplementary Data 3 and Supplementary Fig. 6). Besides peptides resulting from the cleaved flg22 epitope, a flg22 peptide was detected (Fig.6d), suggesting that purified SBT5.2a-His can also release flg22 from the precursor before it is inactivated. Taken together, these data demonstrate that purified SBT5.2a-His preferably cleaves QP2, inactivates flg22 immunogenicity, and quickly degrades flagellin-releasing peptides of the cleaved flg22 epitope.

## SBT5.2 s dampen flg22-induced immune responses

To determine the outcome of SBT5.2 depletion on bacterial growth, we performed infection assays on plants depleted for subtilases with *Pseudomonas syringae*. Bacterial growth on leaves transiently expressing Epi1 is significantly reduced (Fig. 7a), indicating that subtilases might collectively reduce the immunogenicity of flagellin or act in

immune signalling. However, bacterial growth was unaltered in *TRV::SBT5.2* compared to control plants (Fig. 7b), and in *sbt5.2* mutant lines compared to WT plants (Fig. 7c).

The absence of bacterial growth phenotypes on the *sbt5.2* mutant, despite the prolonged stability of flagellin and flg22 in these mutants, prompted us to test if priming by higher flg22 levels can induce immune responses that can be detected in bacterial growth assays. Strong effects on bacterial growth are normally shown by pre-treating (priming) tissues with 1000 nM flg22[23,24]. We reasoned that the effect of SBT5.2 depletion could only be shown at threshold flg22 concentrations because higher concentrations would trigger antibacterial immunity irrespective of SBT5.2. We therefore monitored bacterial growth in leaves pre-treated with 1, 10 and 100 nM flg22. As expected, pre-treatment of WT plants with 100 nM flg22 reduces bacterial

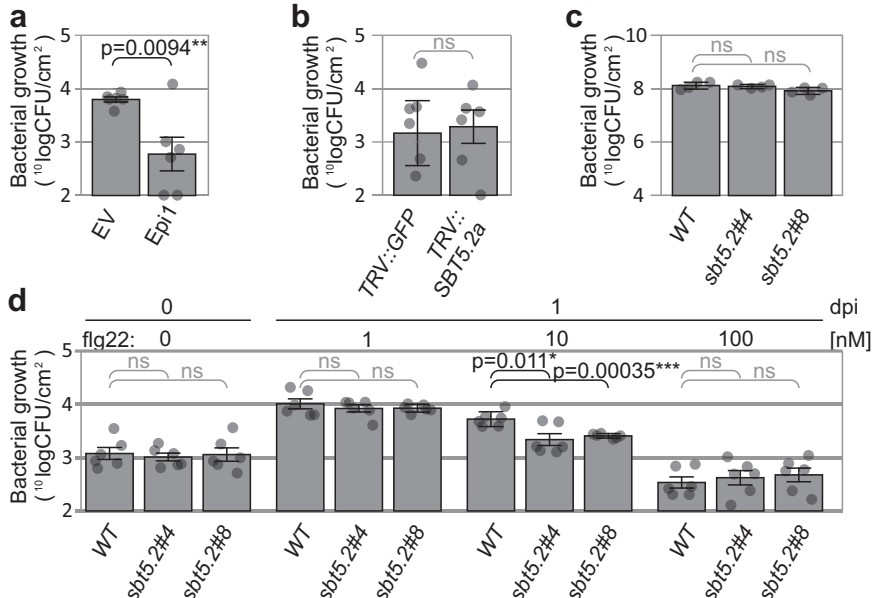

**Fig. 7 | SBT5.2 s dampens elicitor levels to reduce antibacterial immunity.**
**a** Increased antibacterial immunity in leaves is transiently expressing Epi1. Agroinfiltrated leaves expressing empty vector (EV) or Epi1 were sprayed with *Pta*6605 at day 3 post agroinfiltration and bacterial growth was measured three days later. **b** No altered susceptibility *SBT5.2* silenced plants to *Pta*6605. *TRV::GFP* and *TRV::SBT5.2* plants were spray-infected with *Pta*6605 and bacterial growth was determined at 3dpi. **c** No altered susceptibility of *sbt5.2* plants to *Pto*DC3000(*ΔhQ*). Plants were spray inoculated with *Pto*DC3000(*ΔhQ*) and bacterial growth was determined at 3dpi. **d** Immune priming by low flg22 concentrations increases in *sbt5.2* mutant plants. Leaves of 4-week-old WT and *sbt5.2* mutant plants were infiltrated with 1, 10 or 100 nM flg22 or water. After 24 h incubation, the leaves were infiltrated with $1 \times 10^5$ bacteria/ml *Pta*6605. Colony forming units (CFUs) were determined one-day post-infection (dpi). (**a–d**) Bars represent mean, and error bars represent SE of $n = 6$ replicates. The p-value was calculated using the two-sided Student *t* test.

growth compared to pre-treatments with 1 and 10 nM flg22, indicating that these low flg22 concentrations are below the threshold to induce antibacterial immunity (Fig. 7d). In both *sbt5.2* mutants, however, pre-treatment with 10 nM flg22 reduces bacterial growth whereas no altered bacterial growth was detected upon pre-treatment with 1 and 100 nM flg22 (Fig. 7d). Further experiments showed that antibacterial immunity is reduced in *sbt5.2* mutant plants also upon pre-treatment with 5 nM flg22, but not with 20 nM flg22 (Supplementary Fig. 7). These data demonstrate that SBT5.2 dampens antibacterial immunity by inactivating the immunogenicity of flg22 at low flg22 concentrations.

## Discussion

Flagellin and flg22 are almost universally recognised by angiosperm plants. Here, we demonstrated that extracellular SBT5.2 subtilases mediate the quick inactivation of the immunogenicity of both flg22 and flagellin by cleaving in the middle of the flg22 sequence. The increased stability of flg22 in *sbt5.2* mutant plants reduces bacterial growth at low flg22 concentrations, suggesting that SBT5.2 contributes to a spatial-temporal dampening of immunity to restrict the immune responses to the infection site. We also demonstrate that abundant apoplastic subtilases are responsible for these and other flagellin processing events and that polymeric flagellin is protease-resistant.

We demonstrated that flagellin monomers are sensitive to proteolysis in apoplastic fluids in contrast to the flagellin polymers, which are stable. This is consistent with the notion that the discovered processing sites in flagellin monomers are concealed in the polymer. The only solvent-exposed regions of flagellin within the polymer are covered with *O*-glycans, which are thought to protect the polymer against proteases[10,13]. Because the flg22 epitope is concealed within the flagellin polymer, monomeric flagellin is essential for perception. Monomeric flagellin can have different sources. First, monomeric flagellin is produced in the bacterial cell and kept unfolded through its interaction with its chaperone FliS[25]. This pre-polymerised flagellin

could escape into the apoplast, similar to cytoplasmic EF-Tu and CSP proteins that are also perceived in the apoplast[25,26]. Second, some flagellin monomers may fail to polymerise when secreted through the flagellin tubule during flagella synthesis, but most of this leakage is prevented by the flagellin cap protein, FliD[27]. Indeed, the *fliD* mutant of *Pta*6604 secreted large amounts of monomeric flagellin and triggers strong immune responses[28]. Third, flagella are shed upon starvation and during the cell cycle[27,24,29,30], and although flagella do not depolymerise[31], they might release monomers when incubated in an apoplastic environment. Indeed, incubation of flagellin in apoplastic fluids containing BGAL1, which removes the terminal mVio residue from the flagellin glycan, releases more immunogenic flagellin fragments[10], indicating that BGAL1 might promote flagellin depolymerisation.

Our data indicates that cleavage site-1 and site-4 would release immunogenic flagellin fragments from flagellin. N-terminal processing of flg22 at site-1 is suppressed by subtilase inhibitor Epi1 but is not significantly suppressed when silencing *SBT5.2, SBT1.9*, or *SBT1.7 s*. Site-1 processing is, however, significantly reduced in the *sbt5.2* mutant and purified SBT5.2a-His can cleave QP1, indicating that SBT5.2 can process flagellin at the N-terminal site of the flg22 epitope. C-terminal processing at site-4 is suppressed by Epi1 and upon silencing *SBT5.2*. Site-4 processing is also reduced in the *sbt5.2* mutant, although not significantly, and purified SBT5.2a-His can cleave QP4. Taken together, these data indicate that SBT5.2 could cleave flagellin at both ends of flg22 and release flg22 from its precursor. These observations are similar to the recent report that SBT5.2 in Arabidopsis processes at site-4 in the flagellin of *Pseudomonas syringae* pv. *tomato* DC3000 (*Pto*DC3000)[11]. However, we did not detect a two-minute delay in the flagellin-triggered ROS response in the *sbt5.2* mutant, in contrast to the recent study with the Arabidopsis *sbt5.2/sbt1.7* mutant[11].

Besides releasing flg22, our data indicates that flagellin is quickly processed by subtilases cleaving at site-2 and site-3 in the flg22

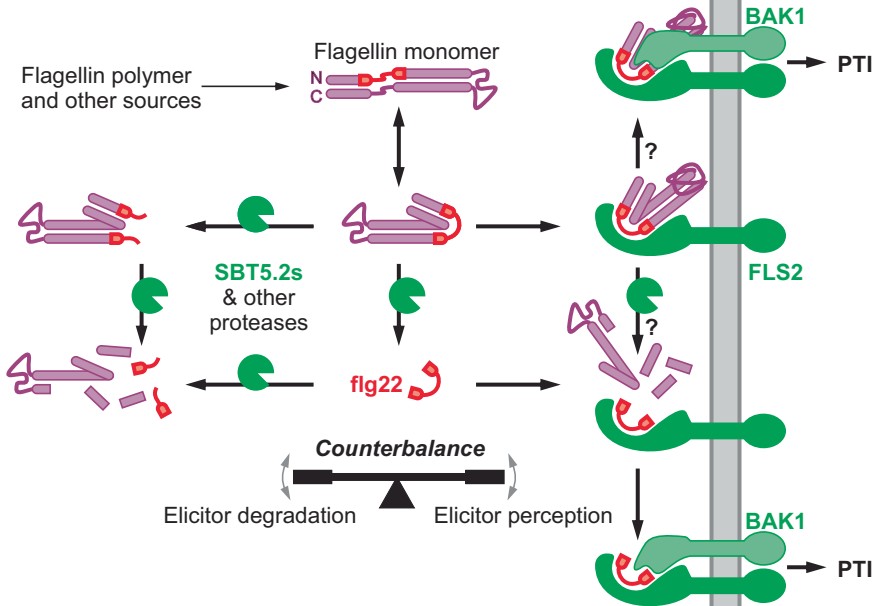

**Fig. 8 | SBT5.2 counterbalances elicitor perception and inactivation.** Elicitor activity of both flagellin and flg22 is quickly inactivated by SBT5.2 s and other proteases cleaving in the middle of the flg22 epitope, yet sufficient flg22 may be released and bind FLS2. Flagellin protein can bind to FLS2 and may associate with BAK1 to trigger immunity without requiring processing. Alternatively, FLS2 may stabilise the flg22 epitope by binding flagellin and SBT5.2 s and other proteases may process flagellin to facilitate BAK1 association and subsequent immune signalling. PTI, pattern-triggered immunity; PM, plasma membrane.

epitope. This is consistent with the fact that we did not detect the full-length flg22 immunogenic peptides in apoplastic fluids with LC-MS/MS. Site-2 (LKIN↓SAKD) is mostly cleaved by SBT5.2 because the processing of QP2 is drastically reduced upon *SBT5.2* silencing and in *sbt5.2* mutants. QP2 is also a very good substrate of purified SBT5.2a-His. Site-3 processing is probably mostly caused by SBT1.9 and SBT1.7a because QP3 processing is significantly reduced upon silencing *SBT1.9* and *SBT1.7a*. QP3 is not efficiently processed by purified SBT5.2a-His, is not significantly reduced in *sbt5.2* mutants, and is not cleaved by purified SBT5.2a-His, indicating that at this site is not preferred by SBT5.2. Processing at site-3 (SAKD↓DAAG) by SBT1.9 is consistent with its annotation as a phytaspase, which cleaves after Asp residues[21]. Interestingly, the recent description of flagellin processing in Arabidopsis seedling exudates indicates that flagellin of *Pto*DC3000 is also processed at site-2 and that this is reduced in seedling exudates of *sbt5.2/sbt1.7* mutants[11], indicating that processing of flagellin within the flg22 epitope might occur universally. Indeed, we detected the same site-2 processing with apoplastic fluids from tomatoes (Supplementary Data 4 and Supplementary Fig. 8), indicating that flagellin processing within the flg22 epitope will occur in many plant species.

Our data indicates that site-2 is preferentially cleaved within flagellin because QP2 is preferentially processed in apoplastic fluids (Fig. 2c) and by purified SBT5.2a-His (Fig. 6a). The preferred processing at site-2 is also supported by modelling of flagellin with SBT5.2a by AlphaFold Multimer (AFM[32]). The best model predicts that the catalytic Ser residue is proximate to the site-2 cleavage site such that the S2 and S4 substrate binding pockets in SBT5.2a are occupied by Ile38 (P2) and Leu36 (P4) of flagellin, respectively (Supplementary Fig. 9). Interestingly, this AFM model indicates that the flagellin monomer could fold on the flg22 hinge to expose this processing site to SBT5.2a (Supplementary Fig. 9).

The quick inactivation of flagellin and flg22 by SBT5.2 cleaving within the flg22 epitope counterbalances the perception of flagellin and flg22 (Fig. 8). The perception of both flagellin and flg22 in *N. benthamiana* indicates that at least some unprocessed flagellin/flg22 must have reached FLS2 and activated PTI signalling. The perception of flagellin/flg22 by FLS2 is possibly mediated by a high affinity for

binding FLS2, indicating that sufficient flagellin/flg22 can bind FLS2 before being cleaved. One caveat in these studies is that the preparation of flagellin monomer includes a heat or low-pH treatment, which might denature flagellin and expose flg22.

Importantly, flagellin protein was found to effectively compete with binding by radiolabeled flg22 (IC50 = 120 nM), in microsomal fractions that presumably lack subtilases[33]. However, although flagellin can bind FLS2, it is unclear at this stage if it would also activate FLS2 signalling because that would require subsequent BAK1 binding to FLS2. The extension of flg22 at its C-terminus is likely to obstruct BAK1 binding, based on the structure of flg22 bound to the FLS2 and BAK1[34]. This indicates that proteases might be required to process flagellin whilst the flg22 epitope is protected by FLS2 (Fig. 8). Indeed, binding of radiolabelled flg22 to cells stabilises the peptide[33], which is consistent with the fact that the cleavage site would be obstructed upon binding FLS2[34]. This process indicates that SBT5.2 might counterbalance flagellin perception by inactivating flagellin before it is perceived, whilst also facilitating flagellin perception when it is bound to FLS2.

Although flg22 and flagellin are more stable in apoplastic fluids of *sbt5.2* plants, they are still slowly degraded. Likewise, the processing of quenched peptides is significantly reduced but not absent in apoplastic fluids of the *sbt5.2* mutant, or in apoplastic fluids from leaves expressing Epi1. These data indicate that proteases other than SBT5.2, or even Epi1-insensitive proteases, contribute to flagellin processing in the apoplast. Indeed, the apoplast contains dozens of active Ser and Cys proteases, as well as aspartic proteases (pepsins) and metalloproteases[15,35]. This suggests that the apoplast is a highly digestive microenvironment, where many proteases collectively contribute to redundant proteolytic machinery that controls levels of immunogenic peptides. The redundancy in the extracellular proteolytic machinery might explain why flagellin is still perceived in the *sbt5.2* mutant. Alternatively, flagellin processing is not required in case BAK1 can assemble with the flagellin-FLS2 complex without flagellin processing. The latter mechanism is similar to how flagellin is recognised on the cell surface receptor by TLR5 in animals[36].

We demonstrated that flg22 and flagellin immunogenicity is prolonged in *sbt5.2* mutants and that this is associated with increased

antibacterial immunity when *sbt5.2* mutants are pre-treated with low flg22 concentrations (5–10 nM flg22). From a plant-centric perspective, these data indicate that SBT5.2 s dampen immune responses both temporally and spatially to avoid the costly indication of antibacterial immunity in tissues that are not or no longer exposed to sufficient bacteria. From a pathogen-centric perspective, bacteria are taking advantage of the plant proteolytic machinery to avoid flg22 accumulation and detection. Relying on plant proteases rather than bacterial-secreted proteases such as AprA[37], has the advantage that flg22 immunogenicity is inactivated beyond the limits of protein diffusion. Interestingly, this mechanism may not be restricted to flagellin perception in *N. benthamiana*, as similar processing of the flg22 epitope has been detected in seedling exudates of Arabidopsis[11] and apoplastic fluids of tomato (Supplementary Fig. 8). This mechanism is probably also universal because the cleavage sites within the flg22 epitope are highly conserved across bacteria (Supplementary Fig. 10). In addition, SBT5.2a is well-conserved across plant species[18] and frequently detected in the apoplast, for instance in Arabidopsis[11]. Given its broad substrate promiscuity, SBT5.2 might be a general protease acting on diverse apoplastic proteins in various plant species, e.g., to maintain protein homoeostasis in the apoplast. The role of SBT5.2 in spatial-temporal control of elicitor levels may also extend to other elicitor proteins. Indeed, we recently found that SBT5.2 is also responsible for reducing levels of csp22 epitope in Cold Shock Proteins (CSPs) of *P. syringae*[22]. Collectively, these data indicate that a conserved extracellular proteolytic machinery consisting of SBT5.2 and other proteases provides spatial-temporal control on elicitor peptide levels to restrict the induction of costly defence responses, whilst simultaneously being used by pathogens to avoid detection.

## Methods

### Plants materials and growth conditions

*Nicotiana benthamiana* plants were grown in a growth chamber at 22 °C and ~60% relative humidity with a 16 h photoperiod and a light intensity of 2000 cd·sr m − 2.

### Flagellin isolation and monomerization

*Pseudomonas syringae* bacteria were grown on LB agar plates for 24 h at 28 °C. Next, bacteria were grown in a 10 ml LB medium supplemented with 10 mM $MgCl_2$ for 24 h at 28 °C. The culture was 1000-fold diluted into 200 ml 10 mM $MgCl_2$ LB and grown for 48 h at 25 °C. The cells were harvested by centrifugation at 5000 × *g* for 10 min at 25 °C, resuspended in 1/3 volume of minimal medium [50 mM potassium phosphate buffer, 7.6 mM $(NH_4)_2SO_4$, 1.7 mM $MgCl_2$, and 1.7 mM NaCl (pH 5.7)] supplemented with 10 mM mannitol, and then incubated for 24 h at 25 °C. After centrifugation at 5000 × *g* for 10 min at 4 °C, the bacterial pellet was resuspended in 1/3 volume of ice-cold PBS pH 7.0. Bacterial flagella were sheared from the cells by vortexing for 1 min (maximum speed) and collected by centrifugation at 5000 × *g* for 20 min at 4 °C. The supernatant was filtered through a 0.45 μm filter on ice. The resultant supernatant was centrifuged at 100,000 × *g* for 2 h at 4 °C. The pellet of purified flagella was suspended in 200 μl of ice-cold PBS pH 7.0. Flagella was further purified using a 100 kDa MWCO. The residues were resuspended in ice-cold PBS pH 7.0. Proteins were quantified using Qubit Fluorometric Quantification and following the manufacturer's instructions (Qubit Protein BR Assay Kit, Thermo Fisher Scientific, UK). Flagella were aliquoted and used immediately or stored at − 80 °C. Flagella were monomerised by heat treatment at 70 °C for 15 min. Purified flagella and flagellin proteins were separated by SDS-PAGE and stained with Coomassie.

### Agroinfiltration

*Agrobacterium tumefaciens* GV3101 (pMP90) was used for agroinfiltration of *N. benthamiana*. Agrobacteria were grown overnight in Luria-Bertani (LB) medium with 10 μg/ml gentamycin and 50 μg/ml kanamycin at 28 °C. For transient expression of proteins in *N. benthamiana*, overnight cultures of Agrobacterium carrying binary vectors listed in Supplementary Table 1 were harvested by centrifugation. Bacterial cells were resuspended in agroinfiltration buffer (10 mM $MgCl_2$, 10 mM MES pH 5.6, 150 μM acetosyringone) and mixed (1:1) with agrobacteria carrying the silencing inhibitor p19 at an $OD_{600}$ of 0.5. After 1 h at 28 °C, cells were infiltrated into leaves of 4-week-old *N. benthamiana*. The infected plants were grown in a growth chamber until use.

### Quenched peptides

The quenched peptides (QPs) from flagellin were commercially synthesised with a DABCYL N-terminal modification and a Glu-EDANS C-terminal modification (GenScript, Piscataway, New Jersey, United States) at a purity of 95% (Supplementary Table 2). They were resuspended in DMSO at a concentration of 1 mM. This stock solution was further diluted in water to a concentration of 200 μM. AFs or purified SBT5.2a were mixed with QPs at a final concentration of 10 μM in a volume of 100 μl, and fluorescence was measured immediately at 21 °C using an Infinite M200 plate reader (Tecan, Mannedorf, Switzerland) every minute over 5 min with an excitation wavelength of 335 nm and emission wavelength of 493 nm.

### Molecular cloning

Used plasmids are summarised in Supplementary Table 1. The binary vector encoding 35S-driven, secreted superfolded GFP with a C-terminal His-tag was produced by cloning synthetic DNA fragment (Supplementary Table 3) into pJK187 with Golden Gate cloning, resulting in pNS205.

### Virus-induced gene silencing (VIGS)

Agrobacterium cultures were grown overnight and resuspended in an agroinfiltration buffer at $OD_{600} = 0.5$. Agrobacteria carrying binary plasmids for expressing RNA2 of TRV-carrying silencing fragments (Supplementary Table 1) were mixed 1:1 with Agrobacteria containing TRV1[38]. After incubation for 1 h at 21 °C, the mixed cultures were infiltrated into the leaves of 14-day-old *N. benthamiana* plants. The infiltrated plants were grown for another 4 to 5 weeks in a growth chamber until use.

### Apoplastic fluids (AFs) isolation

*N. benthamiana* leaves were submerged in ice-cold water and vacuum infiltrated in a 50 ml syringe with a plunger. The surface of water-infiltrated leaves was dried with absorbing paper and leaves were carefully inserted into an empty 20 ml syringe, which was placed in a 50 ml tube. AFs were collected by centrifugation at 2000 × *g* at 4 °C for 10 min. AFs were immediately used or frozen using liquid nitrogen and stored at −20 °C until use.

### SDS-PAGE and Western blot

The protein samples were mixed with 4x gel loading buffer (200 mM Tris-HCl (pH6.8), 400 mM DTT, 8% SDS, 0.4% bromophenol blue, 40% glycerol) and heated at 90 °C for 5 min. The samples were loaded on 15% SDS-PAGE and separated at 180 V in Invitrogen Novex vertical gel tanks. The Amersham® Typhoon was used to measure fluorescence (Cy2: 488 nm for labelled flagellin, Cy5: 685 nm for ladder). The proteins were also visualised by total protein staining with Instant Blue® Coomassie Protein Stain (Abcam ab119211). For western blot analysis, proteins were transferred onto a polyvinylidene difluoride (PVDF, BioRad) membrane using BioRad Trans-blot Turbo® according to the manufacturer's instructions (BioRad Kit 1704275). Blots were blocked for 1 h at 21 °C or overnight at 4 °C with 5% (w/v) skim milk in PBS-T (PBS tablets; Merck 524650, 0.1% Tween-20; Merck P1379). The membrane was incubated with 1:5000 anti-flagellin antibody[39] in 5% (w/v) skim milk for one hour at 21 °C. 1:5000 anti-flagellin antibody was added in the 5% (w/v) skim milk in PBS-T for one hours at 21 °C. Blots were washed twice with PBS-T for 5 min and incubated with 1:5000

HRP-conjugated anti-goat antibody in PBS-T for 1 h at 21 °C. Blots were washed four times with PBS-T for 5 min, and chemiluminescent signals were detected using Clarity Western ECL Substrate (BioRad) and visualised using the ImageQuant® LAS-4000 imager (GE Healthcare, Healthcare Life Sciences, Little Chalfont, UK).

## SBT5.2a-His purification

Four-week-old *N. benthamiana* leaves were infiltrated with a 1:1 mixture (final $OD_{600}$ = 0.5 for each) of Agrobacterium tumefaciens GV3101 containing the silencing suppressor P19 and pPB097[22]. Apoplastic fluid containing SBT5.2a-His was extracted 6 days after infiltration as described above and purified as previously described[40,41]. Briefly, SBT5.2a-His was purified by HisPur™ Ni-NTA resin and concentrated in 25 mM Tris-HCl pH = 6.8 using a 50 kDa MWCO Amicon Ultra-15 filter.

## ROS assays

ROS assays were performed as described previously[10]. Briefly, after incubation in water overnight, one leaf disc (6 mm diameter) was added to 100 μl solution containing 25 ng/μl luminol, 25 ng/μl Horse Radish Peroxidase (HRP) and specified elicitor treatments. For assays with treated elicitors, elicitor peptides or purified flagellin were incubated in AFs from *N. benthamiana* leaves (wild-type, agroinfiltrated, VIGS-silenced or *sbt5.2* mutants) or purified SBT5.2a-HIS for the specified time at 21 °C. After incubation, 25 ng/μl luminol and 25 ng/μl HRP were added to the AFs. Chemiluminescence was measured immediately with the Infinite M200 plate reader (Tecan, Mannedorf, Switzerland) every minute for one hour. Standard errors were calculated at each time point and for each treatment.

## Mass spectrometry

To generate samples for the analysis of endogenously digested peptides, 10 μg/ml of purified flagellin of *Pta*6605 or *Pta*6605*Δfgt1* was incubated for 30 min or 60 min with apoplastic fluids isolated from leaves of *N. benthamiana*, or tomato (Money Maker Cf0), or purified SBT5.2a-HIS at 21 °C (the specific condition is indicated under each figure). The samples were supplemented (AF and/or the purified proteins) with four volumes of MS-grade acetone. After incubating, the mixture was kept on ice for 1 h, and subjected to centrifugation at 18,000 × g for 20 min. Following centrifugation, four-fifths of the supernatants were transferred to fresh Eppendorf tubes, and the acetone was evaporated using vacuum centrifugation. The dried peptide samples were sent to the Analytics Core Facility Essen (ACE) for MS analysis.

## Sample clean-up for LC-MS/MS

Peptides were desalted on homemade C18 StageTips containing two layers of an octadecyl silica membrane (CDS Analytical, Oxford, PA, USA). All centrifugation steps were carried out at room temperature. The StageTips were first activated and equilibrated by passing 50 μL of methanol (600 × g, 2 min), 80% (v/v) acetonitrile (ACN) with 0.5% (v/v) FA (600 × g, 2 min) and 0.5% (v/v) FA (600 × g, 2 min) over the tips. Next, the acidified tryptic digests were passed over the tips (800 × g, 3 min). The immobilised peptides were then washed with 50 μL and 25 μL 0.5% (v/v) FA (800 × g, 3 min). Bound peptides were eluted from the StageTips by application of two rounds of 25 μL 80% (v/v) ACN with 0.5% (v/v) FA (800 × g, 2 min). After elution from the StageTips, the peptide samples were dried using a vacuum concentrator (Eppendorf, Hamburg, Germany) and the peptides were dissolved in 15 μl 0.1% (v/v) FA prior to analysis by MS.

## LC-MS/MS Analysis

LC-MS/MS analysis of peptide samples was performed on an Orbitrap Fusion Lumos mass spectrometer (Thermo Scientific, Waltham, MA, USA) coupled to a Vanquish Neo ultra-high-performance liquid chromatography (UHPLC) system (Thermo Scientific, Waltham, MA, USA) or on a Thermo Orbitrap Elite mass spectrometer coupled to an Easy

nLC 1000 UHPLC. Both systems were operated in the one-column mode. The analytical column was a fused silica capillary (inner diameter 75 μm, outer diameter 360 μm, length 28–35 cm) with an integrated sintered frit packed in-house with Kinetex 1.7 μm XB-C18 core-shell material (Phenomenex, Aschaffenburg, Germany) or Reprosil-Pur 120 C18-AQ,(Dr. Maisch GmbH). The analytical column was encased by a PRSO-V2 column oven (Sonation, Biberach, Germany) and attached to a nanospray flex ion source (Thermo Scientific, Waltham, MA, USA). The column oven temperature was set to 45–50 °C during sample loading and data acquisition. The Vanquish Neo was equipped with two mobile phases: solvent A (2% ACN and 0.2% FA in water) and solvent B (80% ACN and 0.2% FA, in water). The Easy nLC 100 was equipped with solvent A (0.1% FA in water) and solvent B (80% ACN and 0.1% FA in water). All solvents were of UHPLC grade (Honeywell, Charlotte, NC, USA). Peptides were directly loaded onto the analytical column with a maximum flow rate that would not exceed the set pressure limit of 950 bar (usually around 0.5–0.6 μl min⁻¹) and separated on the analytical column by running project-specific gradients. The gradient composition and the MS settings for each experiment can be found in Supplementary Data 5.

## Peptide and protein identification using MaxQuant

RAW spectra were submitted to an Andromeda[42] search in MaxQuant using the default settings[43]. Label-free quantification and match-between-runs was activated[44]. The detailed search settings and which databases were used for the different experiments can be found in Supplementary Data 5.

Briefly, the MS/MS spectra data were searched against project-specific databases. All searches included a contaminants database search (as implemented in MaxQuant, 245 entries). The contaminants database contains known MS contaminants and was included to estimate the level of contamination. Andromeda searches allowed oxidation of methionine residues (16 Da) and acetylation of the protein N-terminus (42 Da) as dynamic modifications. Enzyme specificity was set to "unspecific". The instrument type in Andromeda searches was set to Orbitrap, and the precursor mass tolerance was set to ± 20 ppm (first search) and ± 4.5 ppm (main search). The MS/MS match tolerance was set to ± 0.5 Da. The peptide spectrum matches FDR, and the protein FDR was set to 0.01 (based on the target-decoy approach). For protein quantification unique and razor peptides were allowed. Modified peptides were allowed for quantification. The minimum score for modified peptides was 40. Label-free protein quantification was switched on, and unique and razor peptides were considered for quantification with a minimum ratio count of 2. Retention times were recalibrated based on the built-in nonlinear time-rescaling algorithm. MS/MS identifications were transferred between LC-MS/MS runs with the "match between runs" option in which the maximal match time window was set to 0.7 min and the alignment time window set to 20 min. The quantification is based on the "value at maximum" of the extracted ion current. At least two quantitation events were required for a quantifiable protein. Further analysis and filtering of the results was done in Perseus v1.6.10.0[45]. Comparison of protein group quantities (relative quantification) between different MS runs is based solely on the LFQs as calculated by MaxQuant, MaxLFQ algorithm[44].

## AlphaFold Multimer

All models were generated using AFM v2.1.1[32,46] on the Advanced Research Computing (ARC) clusters as described previously[41]. Briefly, the sequences of mature flagellin and SBT5.2 catalytic domain were submitted to model the structure. Database preset was set to 'full_dbs' to include the full bfd database for multi-sequence alignment search for all predictions. For the flagellin monomer, 'monomer_ptm' was specified for the '--model_preset' flag to obtain the pTM confidence measure. Monomer pTM score was extracted from the.pkl output file specific to the model.

## Phylogenetic analysis

Flagellin sequences were collected for various Pseudomonas strains. Clustal Omega[47] was used for amino acid sequence alignment and neighbour-joining tree construction. The tree was visualised using iTOL[48] and displayed with midpoint rooting.

## Activity-based protein profiling (ABPP)

FP-TAMRA (Thermo-Fisher) was prepared as $10\,\mu M$ stock solutions in dimethyl sulfoxide (DMSO). Serine hydrolase labelling was performed as described previously with minor modifications[49]. Briefly, apoplastic fluids were incubated with $0.2\,\mu M$ FP-TAMRA (Thermo Fischer) in PBS buffer pH7 for 1 h at 21 °C in the dark. Labelling was stopped by adding 4 x gel loading buffer (200 mM Tris-HCl (pH 6.8), 400 mM DTT, 8% SDS, 0.4% bromophenol blue, 40% glycerol) and heating at 90 °C for 5 min. The proteins were separated on SDS-PAGE, and fluorescently labelled proteins were detected by in-gel scanning with the Typhoon FLA 9000 (GE Healthcare Life Sciences) using Cy3 settings (532 nm excitation and 610PB filter).

## Infection assays

flg22 peptides were diluted in water. *N. benthamiana* plants were infiltrated with the different concentrations of flg22 peptide or with water as a mock control. *N. benthamiana* plants were infiltrated when 3-4 weeks old, and 3 fully expanded leaves were infiltrated per plant. 24 h later, infiltrated leaves were infiltrated with $10^5$ CFU/ml *Pta*6605. The next day (1 dpi), three leaf discs were punched with a cork borer from each infected leaf, and surface-sterilised with 15% hydrogen peroxide for 2 min. Leaf discs were then washed twice in MilliQ and dried under sterile conditions. Leaf discs were placed into a 1.5 ml safe-lock Eppendorf tube with three 3 mm diameter metal beads and 1 ml of MilliQ. Tubes were placed in a tissue lyser for five minutes at 30 Hertz/second. 200 µl of the lysed tissue was transferred to the first row (A) of a 96-well plate and then serial 10-fold dilutions were made until the last row (20 µl tissue + 180 µl MilliQ). 20 µl of undiluted tissue and serial dilutions were plated on LB-agar plates containing *Pseudomonas* CFC Agar Supplement (Oxoid SR0103). Plates were allowed to dry, incubated at 28 °C for two days and then colonies were counted.

## Statistics and reproducibility

Statistical tests and the number and type of replicates have been stated in the legends of each figure. All disease assays, ROS assays, QP processing assays and flagellin degradation assays have been reproduced at least once with a similar outcome.

## Reporting summary

Further information on research design is available in the Nature Portfolio Reporting Summary linked to this article.

## Data availability

The proteomics data generated in this study have been deposited in the PRIDE[50] partner repository under accession code PXD056369. The processed proteomics data are available as Supplemental Data 1–4. Source data are provided in this paper.

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

## Acknowledgements

We would like to thank Ben Excel for his help with purifying flagellin; Yuki Ichinose for providing the anti-flagellin antibody; Qiang Cheng for providing the *fls2* mutant, Urszula Pyzio for excellent plant care; and Sarah Rodgers, Patricia Bowman and Caroline O'Brien for excellent technical support. This project was financially supported by BBSRC 17RM3 project BB/R017913/3 'GH35' (P.B.) and 19RM3 project BB/015128/1 'Galactosyrin' (P.B. and N.S.); ERC–2020-AdG project 101019324 'ExtraImmune' (J.H., B.M. and R.H.); BBSRC Interdisciplinary Bioscience DTP project DDT00230 (J.L.); the National Natural Science Foundation of China (No. 32070289 to Y.L.) and the Chinese Academy of Sciences Scholarship (Y.L.).

## Author contributions

P.B. and R.H. conceived the project; P.B. performed most of the experiments with the help of N.S., J.H. and B.M.; F.K. and M.K. performed proteomic analysis; P.B. and J.H. performed experiments with purified SBT5.2; P.B. and D.S. analysed susceptibility of *TRV::SBT5.2* plants; J.L. produced AlphaFold Multimer models; P.B. and Y.L. performed bacterial growth assays on *sbt5.2* plants; P.B. and R.H. wrote the manuscript with input of all authors.

## Competing interests

The authors declare no competing interests.
