## [Peer Review File · Nature Communications]

REVIEWER COMMENTS

Reviewer #1 (Remarks to the Author):

The widely conserved flg22 peptide is one of the most studied elicitors of plant innate immunity, but it is still not known how the epitope is liberated in planta. A recent publication suggested that Subtilisin 5.2 and 1.7 might be involved in C-terminal processing and generation of the active epitope in Arabidopsis (Matsui et al., 2024). The present manuscript demonstrates that both monomeric flagellin and flg22 are quickly inactivated in the apoplast, while polymeric flagellin is stable. Recombinant SBT5.2 cut monomeric flagellin at several sites, including sites that lead to the release of the flg22 epitope. However, detailed biochemical analysis using quenched fluorescence substrates suggests that two sites within the flg22 peptide were kinetically preferred both by recombinant SBT5.2 and, and expression of a subtilisin inhibitor showed, knock-down and knock-out of SBT5.2 in *N. benthamiana* demonstrate that SBT5.2 plays a major role in inactivation of flg22 or monomeric flagellin.

I thoroughly enjoyed the elegant, well-designed and executed experiments that dissect the stability and activity of monomeric, polymeric and processed flagellin in AF and clearly link this to specific proteolytic activities in the apoplast. The apparently contradicting cleavages consistent with release and inactivation of flg22 are resolved in an interesting model that proposed binding of monomeric flagellin to FLS2 might allow subtilisin-mediated cleavage at the activating sites, while preventing the inactivating cleavage. In summary, the paper is well written, provides important novel data and an intriguing hypothesis that may guide future experiments. However, there are a few points that should be addressed:

Please explain how you obtained monomeric and polymeric flagellin already with the description of the first results (Fig 1/ Lines 70/71) - I found this important information only later in line 197 and the legend of Fig 7. Please provide details on the protocol in methods. Can you visualize the oligomerization state on a native gel?

How pure is the preparation of polymeric and monomeric flagellin, respectively? Is it possible that other Sbt 5.2 elicitors are co-purified with monomeric flagellin?

Have the authors tested if the intermediate fragments generated by incubation of purified flagellin with purified SBT5.2 can elicit immune responses? This could be tested by addition of a chemical inhibitor at 10, 30 and 60 min, and assessing responses in leaf discs ROS assay.

Which peptides are generated by incubation of monomeric flagellin with recombinant SBT5.2 (Fig. 6A) or altered after incubation with AF from SBT5.2-knock out plants (Fig. 5B)? MS data on these experiments would help to judge which of the cleavages observed in wt AF are most likely attributed to SBT5.2, and which subsequent processing events may depend on Sbt 5.2 activity beyond the five sites assessed by QF probes. At least for recombinant SBT5.2 this data appears to have been generated (method section line 416), but is not presented.

Please provide lists of peptides identified in mass spectrometry experiments (Figures 2A, S1 and S6) as supplementary material.

Methods:

Details on mass spectrometry data acquisition and description of the software and search settings used for peptide identification are missing. Annotated spectra of the identified peptides should be provided, or preferably the search results and raw data be provided via proteomeXchange partner repository, or similar.

Minor comments

References Chen, Buscaill et al 2024 ... submitted – cite bioRxiv link instead.

Reviewer #2 (Remarks to the Author):

In the present work, Buscaill and colleagues present an analysis of the processing of flagellin into and the subsequent destruction of the immunogenic flg22 epitope that is perceived by most plant species. Through a combination of in vitro biochemical and in planta physiological experiments, they demonstrate that subtilases in the apoplast of *N. benthamiana* can process monomeric flagellin into a collection of derived peptides. Interestingly, SBT5.2 cleaves the immunogenic flg22 epitope internally, rendering it incapable of activating PTI.

This work presents an interesting model where endogenous peptidases can tune immune responses by processing immunogenic peptides. At the end of their study, they propose different models for the perception of flagellin/flg22 by FLS2, integrating their findings on the processing of

flagellin by SBTs. Further information could be provided in support of their model, as detailed in my comments below (see Major comments 1 and 2).

Major comments:

1. Is it possible to perform cleavage experiments on a time scale that would allow the authors to determine which cleavages around the flg22 epitope occur first? For example, is the peptide flanked by site 1 and site 4 released first and subsequently cleaved? Similarly, the authors describe the ordered release of different flagellin fragments (lines 205-211) – could the authors identify the termini of these fragments to provide sequence-level resolution on the cleavage process? Determining which cleavages happen first is important for the model presented at the end of the work where it is speculated that a “sufficient amount of flg22 may be released” to trigger FLS2-mediated signaling.

2. Is flagellin processed by SBT5.2 (or AF) in the presence of the FLS2 ectodomain? To test this, the authors could express and purify ectoFLS2 from benthamiana and perform experiments as depicted in Figure 1. This data would support the conjecture that binding to FLS2 can protect the immunogenic flg22 peptide from subtilase-mediated cleavage.

3. Could the authors provide a LC/MS analysis of a flagellin digest with purified SBT5.2? How similar/different is it from AF digests?

4. Immune priming experiments in Figure 8E and Figure S5: The authors should include mock treatments sampled 1 dpi to match the flg22 treated samples to determine whether the low concentrations of flg22 prime immunity.

Minor comments:

1. Could the authors provide a zoomed-in panel of the flagellin digest that corresponds to the flg22 peptide? This would make it easier for the reader to see the cleavage sites observed in the MS experiment.

2. Where does SBT5.2 cleave elf18?

3. Line 186-187: Figure 6A is described as labelling with an activity probe for Ser hydrolases, but the figure shows an anti-flagellin immunoblot. Please double check that the correct data are presented.

4. Line 247-249: "This is consistent...polymer." – I think part of this sentence is missing.

5. Some of the coomassie stained blots/gels are of poor quality (e.g. Figure 5B and Figure 6A). If better images are available, it would be good to replace these.

6. In Figure 3A, there are data points that fall outside the limits of the Y-axis.

7. In Figure 6 and Figure 7 – the digestion of flagellin does not seem to be consistent. Compare Figure 6A, lane 3 with Figure 7B, lane 7 – these conditions should be identical, if I am not mistaken, but the digestion patterns look different. Is there a specific reason for this, or is this simply variability in the assay?

8. Sometimes enzyme concentrations are reported as ng/ul and sometimes as ug/ml. Please choose one for consistency.

9. How confident are the authors in the purity of the SBT5.2 – it would be nice in one assay with a control (e.g. catalytically dead SBT) purified in parallel. Can the authors be confident that all the bands in Fig S3 are isoforms of SBT5.2, not co-purified subtilases.

10. Line 205-208 The determination of sizes from the blot is challenging – please add an ~ to indicate this is an approximation.

11. Do the authors have any speculations about specificity. Could other related SBTs fulfil the same function, but are just not expressed sufficiently in leaf tissues?

12. There seems to be a discrepancy between the Qp1 and Qp4 cleavage activity between AF from TRV::SBT5.2 and sbt5.2 CRISPR plants (Fig 4A vs Fig 5D). How do the authors explain these differences.

Reviewer #3 (Remarks to the Author):

The authors report the identification of the *Nicotiana benthamiana* subtilase SBT5.2 as a processing enzyme for bacterial flg22. A very recent work on *Arabidopsis* revealed that SBT5.2 and the related SBT1.7 could release signaling active flg22 from flagellin. The authors now suggest that *N. benthamiana* SBT5.2's primary function is the degradation of flg22 for signal clearance to avoid over activation of immunity. This work is a nice complementation to the previous *Nature Communications* paper by Matsui and colleagues, but proposes that that the previously reported flg22 release by SBT5.2 is rather a secondary function. Their conclusions are supported by a combination of biochemical, genetic and physiological experiments. It is a very nice and convincing paper, further substantiating the ability of subtilases to process flagellin. In light of the recent Matsui et al. paper (*Nat. Comm.* 2024), SBT5.2's primary function remains under debate (degradation/release of flg22). I would like to see some more physiological experiments demonstrating the importance of the observed cleavage/processing for the support or inhibition of FLS2-triggered immunity. This remains the weakest point of the presented manuscript (but also of the manuscript by Matsui and colleagues).

Main points:

- Given the proposed inhibited degradation of flagellin/flg22 in *sbt5.2* mutants, I would like to see a ROS burst assay comparing flagellin/flg22 treatment in WT and *sbt5.2* mutants, including careful interpretation, depending on the outcome. In particular, Matsui et al. report a delayed ROS burst response in *sbt5.2/sbt1.7* mutants upon flagellin treatment, indicative of delayed flg22 release. The authors should test if similar phenomena occur in *N. benthamiana* as well.
- In a similar context, I would like to see sensitivity of *sbt5.2* plants to other flagellin/flg22-induced immune outputs. In particular, late responses might provide insight into altered flagellin stability, e.g. gene expression or seedling growth inhibition
- Similar to the presented flg22-induced resistance experiment, physiological assays testing SBT5.2's effect on PTI responses may require dose response experiments using different concentrations of flg22. I encourage the authors to do so.
- Since SBT5.2 can also degrade elf18 and csp22, I encourage the authors to similarly test sensitivity of *sbt* mutants to csp22 (elf18 does not work since the receptor is missing in *N. benth.*).

Minor:

- Figure 4B would greatly benefit from an additional control other than TRV::GFP. Fig 4A shows that QP2 processing (which would cleave flg22) is unaffected by SBT1.9/SBT1.7 silencing. Please use either of those as a better control.

- Fig 4A: QP2 processing in TRV::SBT1.7a seems to not differ a lot from 100% Processing, yet is indicated as strongly significant. Please carefully check this. Also, the text does not indicate that QP2 processing depends on SBT1.7a

We like to thank the reviewers for their critical comments, which we have used to improve the manuscript, as detailed below. In addition to the changes in response to the comments from the three reviewers, we have explained the flagellin purification and monomerization in more detail in the results section and reorganised the order of the subfigures to improve the consistence and flow of the manuscript.

Reviewer #1 (Remarks to the Author):

The widely conserved flg22 peptide is one of the most studied elicitors of plant innate immunity, but it is still not known how the epitope is liberated in planta. A recent publication suggested that Subtilisin 5.2 and 1.7 might be involved in C-terminal processing and generation of the active epitope in Arabidopsis (Matsui et al., 2024). The present manuscript demonstrates that both monomeric flagellin and flg22 are quickly inactivated in the apoplast, while polymeric flagellin is stable. Recombinant SBT5.2 cut monomeric flagellin at several sites, including sites that lead to the release the flg22 epitope. However, detailed biochemical analysis using quenched fluorescence substrates suggests that two sites within the flg22 peptide were kinetically preferred both by recombinant SBT5.2 and, and expression of a subtilisin inhibitor showed, knock-down and knock-out of SBT5.2 in *N. benthamiana* demonstrate that SBT5.2 plays a major role inactivation of flg22 or monomeric flagellin.

I thoroughly enjoyed the elegant, well-designed and executed experiments that dissect the stability and activity of monomeric, polymeric and processed flagellin in AF and clearly link this to specific proteolytic activities in the apoplast. The apparently contradicting cleavages consistent with release and inactivation of flg22 are resolved in an interesting model that proposed binding of monomeric flagellin to FLS2 might allow subtilisin-mediated cleavage at the activating sites, while preventing the inactivating cleavage. In summary, the paper is well written, provides important novel data and an intriguing hypothesis that may guide future experiments. However, there are a few points that should be addressed:

Please explain how you obtained monomeric and polymeric flagellin already with the description of the first results (Fig 1/ Lines 70/71) - I found this important information only later in line 197 and the legend of Fig 7. Please provide details on the protocol in methods. Can you visualize the oligomerization state on a native gel? **RESPONSE: The reviewer is right. We have added a *NEW Fig. 1a* (originally Fig. 7a) showing that polymeric flagellin cannot pass the 100kDa MWCO filter, but monomeric flagellin can. We repeated assays with these MWCO fractions and show in *NEW Fig. 1b* that polymeric flagellin is not immunogenic but monomeric flagellin is. We have also added the description of flagellin isolation to the Results and the Methods. The flagellin polymer is too large to separate on a native gel but we were able to demonstrate its polymeric nature by the fact that it is unable to pass through the 100 kDa MWCO filter.**

How pure is the preparation of polymeric and monomeric flagellin, respectively? Is it possible that other Sbt 5.2 elicitors are co-purified with monomeric flagellin? **RESPONSE: We have moved the first half of the original Figure 7A to *NEW Fig. 1a* do show the purify of the purified flagellin on a Coomassie gel. We could not observe any response when the monomerised flagellin on the *fls2* mutant (*NEW Fig. 1c*), demonstrating that no other elicitors were co-purified with monomeric flagellin.**

Have the authors tested if the intermediate fragments generated by incubation of purified flagellin with purified SBT5.2 can elicit immune responses? This could be tested by addition

of a chemical inhibitor at 10, 30 and 60 min, and assessing responses in leaf discs ROS assay. **RESPONSE:** This assay is not so different from testing immunogenicity of flagellin incubated with AF for different time points, which we have provided in **Fig. 1e**. Protease inhibitors may stop further processing but this is irrelevant when added to leaf discs that immediately respond. Contrary, the protease inhibitors may also affect signalling. Determining the immunogenicity of each intermediate is challenging with these experiments because multiple intermediates co-exist at the different timepoints.

Which peptides are generated by incubation of monomeric flagellin with recombinant SBT5.2 (Fig. 6A) or altered after incubation with AF from SBT5.2-knock out plants (Fig. 5B)? MS data on these experiments would help to judge which of the cleavages observed in wt AF are most likely attributed to SBT5.2, and which subsequent processing events may depend on Sbt 5.2 activity beyond the five sites assessed by QF probes. At least for recombinant SBT5.2 this data appears to have been generated (method section line 416), but is not presented. **RESPONSE:** We have repeated peptide release from flagellin by purified SBT5.2 -His and included purified GFP-His as a negative control for more replicates (n=3) at t=30'. **NEW Fig. 6d** demonstrates that purified SBT5.2a-His releases the same fragments from flagellin as detected in apoplastic fluids (**Fig. 2a**), except for the fragments generated by phytaspase cleaving site-3. The exopeptidase activity detected in this assay indicates that SBT5.2a-His might have both endo and exopeptidase activity. The few peptides detected in the GFP-His control (**NEW Fig. S6**) were also detected upon incubation in water, suggesting that these peptides were present in flagellin purified from Pta6605.

Please provide lists of peptides identified in mass spectrometry experiments (Figures 2A, S1 and S6) as supplementary material. **RESPONSE:** We have added these data as **NEW Supplementary Tables S1, S2, S3 and S4**.

Methods:

Details on mass spectrometry data acquisition and description of the software and search settings used for peptide identification are missing. Annotated spectra of the identified peptides should be provided, or preferably the search results and raw data be provided via proteomeXchange partner repository, or similar. **RESPONSE:** We have added details of the MS data acquisition and spectral annotation to the M&M section and have uploaded the datafiles to PRIDE, with the accession codes provided in the data availability section.

Minor comments

References Chen, Buscaill et al 2024 ... submitted – cite bioRxiv link instead. **RESPONSE:** This is now published in *Nature Plants* and has been cited accordingly.

Reviewer #2 (Remarks to the Author):

In the present work, Buscaill and colleagues present an analysis of the processing of flagellin into and the subsequent destruction of the immunogenic flg22 epitope that is perceived by most plant species. Through a combination of in vitro biochemical and in planta physiological experiments, they demonstrate that subtilases in the apoplast of *N. benthamiana* can process monomeric flagellin into a collection of derived peptides. Interestingly, SBT5.2 cleaves the immunogenic flg22 epitope internally, rendering it incapable of activating PTI.

This work presents an interesting model where endogenous peptidases can tune immune responses by processing immunogenic peptides. At the end of their study, the propose different models for the perception of flagellin/flg22 by FLS2, integrating their findings on the processing of flagellin by SBTs. Further information could be provided in support of their model, as detailed in my comments below (see Major comments 1 and 2).

Major comments:

1. Is it possible to perform cleavage experiments on a time scale that would allow the authors to determine which cleavages around the flg22 epitope occur first? For example, is the peptide flanked by site 1 and site 4 released first and subsequently cleaved? Similarly, the authors describe the ordered release of different flagellin fragments (lines 205-211) – could the authors identify the termini of these fragments to provide sequence-level resolution on the cleavage process? Determining which cleavages happen first is important for the model presented at the end of the work where it is speculated that a “sufficient amount of flg22 may be released” to trigger FLS2-mediated signaling. **RESPONSE:** We have isolated the intermediates from gel, digested them with trypsin and analysed them by LC-MS/MS, but found no differences between the gel slices, indicating that the intermediates did not separate sufficiently to distinguish them by MS. We have also monitored the release of flagellin fragments with purified SBT5.2a-His over time (10', 30', 60') but did not detect a clear difference between the time points, probably because the processing could not be synchronised sufficiently because these assays contain a flagellin overdose. We did, however, detect the flg22 peptide when flagellin was incubated with purified SBT5.2a-His in addition to its processing products (**NEW Fig. 6d**). Although QP processing and AFM modelling suggests that there is a preference for cleaving within the flg22 epitope, and SBT1.9 would cleave flg22 at site-3 in AF, we are not able to claim that this site is always cleaved first.

2. Is flagellin processed by SBT5.2 (or AF) in the presence of the FLS2 ectodomain? To test this, the authors could express and purify ectoFLS2 from benthamiana and perform experiments as depicted in Figure 1. This data would support the conjecture that binding to FLS2 can protect the immunogenic flg22 peptide from subtilase-mediated cleavage. **RESPONSE:** Upon this request, we have cloned and transiently expressed the ectodomain of *NbFLS2* by agroinfiltration but could not detect this product by western blot or Coomassie. We could therefore not perform the suggested experiment. However, the evidence that the flg22 epitope is protected upon binding is provided in Figure 4 from Meindl et al., *Plant Cell* 2000 (copied below). This evidence, together with the fact that flagellin itself competes for flg22 binding in Figure 6 of the same publication (copied below) and in Figure 4 of Bauer et al, *JBC* 2001 (copied below), supports our hypothesis that FLS2 might stabilise the flg22 epitope upon flagellin binding.

Figure 4. Stability of Radioligand ^{125}I -Tyr-flg22 in Cell Suspension at Room Temperature.

●	fig22	QRLSTGSRINSAKDDAAGLQIA
◇	fig15	RINSAKDDAAGLQIA
□	fig8	DAAGLQIA
◆	fig15-Δ7	RINSAKDD
▼	fig15 (R. melliloti)	RVGQAADNAAWWSIA
△	flagellin	32-kD protein

Figure 6. Competition of ^{125}I -Tyr-flg22 Binding by Flagellin and Different Peptides.

FIG. 5. Correlation of biological activities and binding affinities for flagellin and flagellin-derived peptides. Relative activities

3. Could the authors provide a LC/MS analysis of a flagellin digest with purified SBT5.2? How similar/different is it from AF digests? **RESPONSE:** We have repeated this experiment with n=3 replicates with both purified SBT5.2a-His and GFP-His and added these data as **NEW Fig. 56d and S6**. Processing by purified SBT5.2a-His is remarkably similar, although processing at site-3 is absent, consistent by this being done by SBT1.9/1.7. The fact that the

peptides are still staggered at both termini indicates that SBT5.2a-His might also have exopeptidase activity.

4. Immune priming experiments in Figure 8E and Figure S5: The authors should include mock treatments sampled 1 dpi to match the flg22 treated samples to determine whether the low concentrations of flg22 prime immunity. **RESPONSE:** We have not performed these experiments because we provided data from which we can conclude that priming works at 5, 10 and 100 nM when compared to 1 nM. We are not making any claim for immune priming that might be triggered by 1 nM flg22 even though we do see reduced bacterial growth in *sbt5.2* mutants pre-treated with 1 nM flg22, but this is not significant.

Minor comments:

1. Could the authors provide a zoomed-in panel of the flagellin digest that corresponds to the flg22 peptide? This would make it easier for the reader to see the cleavage sites observed in the MS experiment. **RESPONSE:** We have added a zoomed-in panel to all MS experiments.

2. Where does SBT5.2 cleave elf18? **RESPONSE:** We removed this experiment because the degradation of elf18 in *N. benthamiana* makes no biological sense in solanaceous plants because these plants lack the EFR receptor, as flagged by reviewer-3.

3. Line 186-187: Figure 6A is described as labelling with an activity probe for Ser hydrolases, but the figure shows an anti-flagellin immunoblot. Please double check that the correct data are presented. **RESPONSE:** Thanks for spotting this error. We now refer to the correct Supplemental Figure.

4. Line 247-249: “This is consistent...polymer.” – I think part of this sentence is missing. **RESPONSE:** We have revised this sentence.

5. Some of the Coomassie stained blots/gels are of poor quality (e.g. Figure 5B and Figure 6A). If better images are available, it would be good to replace these. **RESPONSE:** We have adjusted the contrast of these images.

6. In Figure 3A, there are data points that fall outside the limits of the Y-axis. **RESPONSE:** This figure has been adjusted by extending the Y-axis.

7. In Figure 6 and Figure 7 – the digestion of flagellin does not seem to be consistent. Compare Figure 6A, lane 3 with Figure 7B, lane 7 – these conditions should be identical, if I am not mistaken, but the digestion patterns look different. Is there a specific reason for this, or is this simply variability in the assay? **RESPONSE:** In contrast to the time course of the original Fig. 7b, where samples were immediately boiled in GLB, samples of the original Fig. 7a were first passed over a MWCO filter, which adds 40 minutes to the incubation times. Also, samples shown in the original Fig. 6 were incubated for 40 minutes. We have corrected this in the legend and figure. We have also removed the incubation of the monomer/polymer with purified SBT5.2 (original Fig. 7a, right half) because the stability of the polymer was already discussed in the beginning of the manuscript and this experiment does not add much to this observation.

8. Sometimes enzyme concentrations are reported as ng/ul and sometimes as ug/ml. Please choose one for consistency. **RESPONSE:** Good point. All concentrations are now expressed in µg/ml.

9. How confident are the authors in the purity of the SBT5.2 – it would be nice in one assay with a control (e.g. catalytically dead SBT) purified in parallel. Can the authors be confident that all the bands in Fig S3 are isoforms of SBT5.2, not co-purified subtilases. **RESPONSE:** Catalytic mutants of subtilases do not accumulate upon transient expression but we have included GFP-His purified from AF of agroinfiltrated plants as a negative control. The absence of flagellin processing in the GFP-His control, and site-3 processing in the SBT5.2a-His sample demonstrates that SBT5.2a-His was not containing co-purified subtilases.

10. Line 205-208 The determination of sizes from the blot is challenging – please add an ~ to indicate this is an approximation. **RESPONSE:** We have made this revision

11. Do the authors have any speculations about specificity. Could other related SBTs fulfil the same function, but are just not expressed sufficiently in leaf tissues? **RESPONSE:** This topic remains to be explored. The fact that flagellin processing is similar in AF of tomato and Arabidopsis indicates that flg22 inactivation is a general phenomenon. We have extended this topic in the discussion.

12. There seems to be a discrepancy between the Qp1 and Qp4 cleavage activity between AF from TRV::SBT5.2 and *sbt5.2* CRISPR plants (Fig 4A vs Fig 5D). How do the authors explain these differences. **RESPONSE:** QP1 is cleaved by purified SBT5.2 and is significantly less cleaved in AF of the *sbt5.2* mutant. We believe that the detection of unaltered QP1 cleavage in AF of TRV::SBT5.2 plants is caused by one aberrant datapoint caused by the low number of replicates (n=3) and variation in VIGS efficiency. Processing of QP4 in AF of *sbt5.2* mutants was reduced but not significantly, which correlates with a low intensity of the QP4 fluorophore. We noticed this when dissolving the peptide powder. We tried to repeat these assays with newly synthesised QP4 but the company was unable to repeat its synthesis, indicating that QP4 solubility is problematic.

Reviewer #3 (Remarks to the Author):

The authors report the identification of the *Nicotiana benthamiana* subtilase SBT5.2 as a processing enzyme for bacterial flg22. A very recent work on Arabidopsis revealed that SBT5.2 and the related SBT1.7 could release signaling active flg22 from flagellin. The authors now suggest that *N. benthamiana* SBT5.2's primary function is the degradation of flg22 for signal clearance to avoid over activation of immunity. This work is a nice complementation to the previous Nature Communications paper by Matsui and colleagues, but proposes that that the previously reported flg22 release by SBT5.2 is rather a secondary function. Their conclusions are supported by a combination of biochemical, genetic and physiological experiments. It is a very nice and convincing paper, further substantiating the ability of subtilases to process flagellin. In light of the recent Matsui et al. paper (Nat. Comm. 2024), SBT5.2's primary function remains under debate (degradation/release of flg22). I would like to see some more physiological experiments demonstrating the importance of the observed cleavage/processing for the support or inhibition of FLS2-triggered immunity. This remains the weakest point of the presented manuscript (but also of the manuscript by Matsui and colleagues).

Main points:

- Given the proposed inhibited degradation of flagellin/flg22 in *sbt5.2* mutants, I would like to see a ROS burst assay comparing flagellin/flg22 treatment in WT and *sbt5.2* mutants,

including careful interpretation, depending on the outcome. In particular, Matsui et al. report a delayed ROS burst response in *sbt5.2/sbt1.7* mutants upon flagellin treatment, indicative of delayed flg22 release. The authors should test if similar phenomena occur in *N. benthamiana* as well. **RESPONSE:** We have added the requested experiments and found no difference in the ROS response peak times for monomerised flagellin between WT and *sbt5.2* mutants, unlike the previous study in *Arabidopsis sbt5.2/1.7* mutants (**NEW Fig. 6e and 6f**).

- In a similar context, I would like to see sensitivity of *sbt5.2* plants to other flagellin/flg22-induced immune outputs. In particular, late responses might provide insight into altered flagellin stability, e.g. gene expression or seedling growth inhibition. **RESPONSE:** We thank the reviewer for this suggestion, but we already provide bacterial growth assays showing that bacteria grow less in *sbt5.2* mutants upon priming with low flg22 concentrations. Besides ROS assays, which are widely used in the field, this is the ultimate proof that SBT5.2 dampens immune responses by degrading flg22, which is the key message of this manuscript.

- Similar to the presented flg22-induced resistance experiment, physiological assays testing SBT5.2's effect on PTI responses may require dose response experiments using different concentrations of flg22. I encourage the authors to do so. **RESPONSE:** We have provided ROS assays at different flg22 concentrations in **NEW Fig. 5g and S3**. We have provided a large amount of data to support the key conclusion of the manuscript that flagellin and flg22 are inactivated by SBT5.2 cleaving in the flg22 epitope.

- Since SBT5.2 can also degrade *elf18* and *csp22*, I encourage the authors to similarly test sensitivity of *sbt* mutants to *csp22* (*elf18* does not work since the receptor is missing in *N. benth.*). **RESPONSE:** We have removed the *elf18* data because *N. benthamiana* lacks EFR. The processing and perception of *csp22* has been studied in detail (Chen et al., *Nat. Plants* 2024). As with flg22, ROS responses induced by *csp22* are identical between WT and *sbt5.2* plants (see FigS4 in Chen et al. 2024, copied below).

Minor:

- Figure 4B would greatly benefit from an additional control other than TRV::GFP. Fig 4A

shows that QP2 processing (which would cleave flg22) is unaffected by SBT1.9/SBT1.7 silencing. Please use either of those as a better control. **RESPONSE:** We do not see why *TRV::GFP* is not a good negative control. The argument to focus on SBT5.2 is given from the QP processing in *TRV::SBT5.2* plants, and is later confirmed with the *sbt5.2* mutants, which is the focus of this manuscript. We feel that this narrative is correct and made some changes in the text to stress this narrative.

- Fig 4A: QP2 processing in *TRV::SBT1.7a* seems to not differ a lot from 100% Processing, yet is indicated as strongly significant. Please carefully check this. Also, the text does not indicate that QP2 processing depends on SBT1.7a. **RESPONSE:** This is caused by using only n=3 replicates. We have added a sentence to the legend of Fig4A to explain that the significance of this differential is probably not relevant, given the high values.

REVIEWERS' COMMENTS

Reviewer #1 (Remarks to the Author):

The reviewers have adequately addressed all of my concerns. I particularly appreciate the rearranged figures and clear additional data presented in Figures 1 and 6, as well as the more detailed information on the flagellin purification and mass spectrometry experiments.

Congratulations on a very nice work.

Reviewer #2 (Remarks to the Author):

The authors have satisfactorily addressed my comments.

Reviewer #3 (Remarks to the Author):

I have reviewed the manuscript before. The authors addressed the raised concerns sufficiently and I can now recommend publication of the article.